# The TropoPause Composition TOwed Sensor Shuttle (TPC-TOSS): A new airborne dual platform approach for atmospheric composition measurements at the tropopause

Heiko Bozem<sup>1</sup>, Philipp Joppe<sup>1,2</sup>, Yun Li<sup>3</sup>, Nicolas Emig<sup>1</sup>, Armin Afchine<sup>4</sup>, Anna Breuninger<sup>5</sup>, Joachim Curtius<sup>5</sup>, Stefan Hofmann<sup>8</sup>, Sadath Ismayil<sup>7</sup>, Konrad Kandler<sup>7</sup>, Daniel Kunkel<sup>1</sup>, Arthur Kutschka<sup>8</sup>, Hans-Christoph Lachnitt<sup>1</sup>, Andreas Petzold<sup>3,6</sup>, Sarah Richter<sup>5</sup>, Timo Röschenthaler<sup>8</sup>, Christian Rolf<sup>4</sup>, Lisa Schneider<sup>7</sup>, Johannes Schneider<sup>2</sup>, Alexander Vogel<sup>5</sup>, and Peter Hoor<sup>1</sup>

Correspondence: Heiko Bozem (bozemh@uni-mainz.de)

Abstract. In this paper we introduce the new TropoPause Composition TOwed Sensor Shuttle (TPC-TOSS), which constitutes an advanced development of the AIRcraft TOwed Sensor Shuttle (AIRTOSS), introduced by Frey et al. (2009). As part of a tandem measurement platform with a Learjet 35A, both platforms were equipped with redundant instruments for co-located measurements of aerosol size distribution (Ultra-High Sensitivity Aerosol Spectrometer, UHSAS), ozone (2BTech model 205), cloud particles (Back-Scatter Cloud Probe, BCP), as well as relative humidity, temperature and pressure. To measure the exact position of the two platforms as well as the relative distance of the TPC-TOSS to the Learjet a Global Positioning System (GPS) is installed on both platforms. Two identical Inertial Navigation Systems (INS) further allow to monitor attitude angles (roll, pitch, and heading) and accelerations.

Laboratory tests before and ground tests as well as inflight tests during the intensive operation period show a good agreement of the ozone and temperature measurements of better than 4.2 ppbv + 1.1 % (ozone) and 0.5 °C (temperature) at a noise level of  $\pm$  (2 ppbv + 0.5 %) for 2 s data (ozone) and  $\pm$  0.1 K for 1 Hz data (temperature). Stability of the ozone monitor mounted in the TPC-TOSS has been tested and is estimated to be 2.2 ppbv (offset, 1  $\sigma$ ) and 0.7 % (gain, 1  $\sigma$ ), respectively, based on the drift of offset and gain during regular calibrations between measurement flights in the two weeks operation period.

The new TPC-TOSS was successfully flown during the TPEx I (TropoPause composition gradients and mixing Experiment) mission in June 2024 and performed four flights covering the altitude range between 6.4 and 10.9 km. The tropopause was crossed several times as evident from different temperature and ozone gradients as well as gradients of the aerosol number concentration. With the setup we are able to resolve transient stability and composition gradients ranging from almost zero or even negative to strong positive gradients of up to 25 K km<sup>-1</sup> for potential temperature and from inverted to strong positive

<sup>&</sup>lt;sup>1</sup>Johannes Gutenberg University of Mainz, Institute for Atmospheric Physics, Mainz, Germany

<sup>&</sup>lt;sup>2</sup>Aerosol Chemistry Department, Max Planck Institute for Chemistry, Mainz, Germany

<sup>&</sup>lt;sup>3</sup>Institute of Climate and Energy Systems 3 – Troposphere, Forschungszentrum Jülich GmbH, Jülich, Germany

<sup>&</sup>lt;sup>4</sup>Institute of Climate and Energy Systems 4 – Stratosphere, Forschungszentrum Jülich GmbH, Jülich, Germany

<sup>&</sup>lt;sup>5</sup>Institute for Atmospheric and Environmental Sciences, Goethe University Frankfurt, Frankfurt am Main, Germany

<sup>&</sup>lt;sup>6</sup>Institute for Atmospheric and Environmental Research, University of Wuppertal, Wuppertal, Germany

<sup>&</sup>lt;sup>7</sup>Institute for Applied Geoscience, Technical University Darmstadt, Darmstadt, Germany

<sup>&</sup>lt;sup>8</sup>enviscope GmbH, Frankfurt, Germany

vertical gradients of ozone of up to  $800 \text{ ppbv km}^{-1}$ , respectively. These gradients are caused by transport and mixing due to convection or shear induced turbulence at the tropopause.

## 1 Introduction

The tropopause is naturally defined by the change of the vertical temperature gradient from the troposphere and a mostly moist adiabatic temperature lapse rate to neutral or positive temperature gradients due to increased shortwave absorption from ozone production in the stratosphere. According to the World Meteorological Organization (WMO, 1957), the tropopause can be defined as the lowest level the temperature lapse rate does not exceed 2 K km<sup>-1</sup> and stays on average below this value for any layer between this altitude and any level above within the next 2 km. As a consequence the emerging increase of static stability makes the thermal tropopause a transport barrier, which in turn leads to strong gradients of tracers (e.g. Bethan et al., 1996; Hoor et al., 2002; Pan et al., 2004; Bauchinger et al., 2025). In the extratropics, the tropopause location is highly variable in time and space being linked to the synoptic conditions. Further non-conservative (diabatic) processes modify composition gradients and in turn the tropopause location itself. The representation of tropopause gradients is, however, crucial for understanding and quantifying the climate impact of radiatively active substances like water vapor, ozone, ice and aerosol particles (e.g. Randel et al., 2007; Fusina and Spichtinger, 2010). These composition gradients are highly variable as a result of the aforementioned variability of the tropopause as well as mixing processes associated with small scale (and large scale) diabatics (Kunkel et al., 2019; Lachnitt et al., 2023).

Measurements of these composition gradients at small scales are difficult to achieve: remote sensing methods suffer from limited resolution due to vertical or horizontal averaging kernels, vertical soundings only provide single profiles while aircraft measurements deliver data just along the flight trajectory. Highly transient phenomena like turbulent mixing processes or occurrence of cirrus clouds, overshooting anvil tops, etc., cannot be covered by horizontal flight tracks on different legs, since the relevant features may disappear when performing stacked level flights. One approach in former studies was to perform co-located measurements with two aircraft, for example during CRYSTAL-FACE (Cirrus Regional Study of Tropical Anvils and Cirrus Layers – Florida Area Cirrus Experiment) (Jensen et al., 2004) or with the *Polar 5* and *Polar 6* aircraft during ACLOUD (Ehrlich et al., 2019). The *Polar* aircraft are almost identical making them well suited for co-located measurements as shown by Maherndl et al. (2024). However, flights with *Polar 5* and *Polar 6* aircraft could only be performed below 5 km altitude due to aircraft performance capabilities. In general, coordinated measurements involving two different aircraft often suffer from difficulties of exact horizontal co-location at two different altitudes due to different aircraft speeds, as pointed out by Klingebiel et al. (2017, and references therein). Further, measurements at a vertical distance of just 100–200 m are difficult to realize with aircraft for safety reasons.

Simultaneous measurements of such small scale structures may deliver novel information on the effect of transient dynamical processes and their impact on species gradients at the tropopause. In earlier studies, towed sensors were introduced for colocated measurements. There are only very few setups of these devices available. Recently, the Alfred Wegener Institute (AWI) and the Leibniz Institute for Tropospheric Research (TROPOS) developed the T-Bird (Jurányi et al., 2025), a sensor shuttle

for turbulence, aerosol and trace species measurements in the lowermost Arctic boundary layer towed by AWIs *Polar* aircraft. Besides this, only helicopter based dual platform designs are available. Both, ACTOS (Airborne Cloud Turbulence Observation System) (Siebert et al., 2006) and SMART-HELIOS (HELIcopter-borne Observations of Spectral Radiation) (Werner et al., 2013, 2014) as well as HELiPOD (Pätzold et al., 2023) are instrument platforms developed to be towed by a helicopter for measurements of boundary layer characteristics with respect to clouds and chemical composition or solar spectral reflectivity. To the best of our knowledge, such an approach has not yet been applied to tropopause altitudes or tropopause-relevant composition measurements.

Here, we present a novel development, which builds on previous experiences for a radiation / cirrus payload (Frey et al., 2009, 2014; Finger et al., 2016; Klingebiel et al., 2017). The new setup of the TPC-TOSS (TropoPause Composition TOwed Sensor Shuttle) includes measurements of ozone, GPS information, aerosol size distribution from 95–1000 nm as well as sensors for humidity and temperature, which are operated simultaneously on the Learjet.

This approach addresses the challenge of characterizing transient fine-scale structures and composition variability in the tropopause region. Such features are particularly associated with composition gradients and variations in temperature, humidity and ozone. To resolve these structures, simultaneous measurements typically require a vertical resolution of 100–150 m and accuracies better than about 0.2 K for temperature and a few percent for humidity and trace species measurements.

Notably, conventional in situ single-platform approaches cannot provide truly simultaneous observations of transient structures with a lifetime of less than a few minutes. The dual-platform concept combining TPC-TOSS and the Learjet directly addresses this gap by deploying two synchronized payloads at slightly different altitudes. This configuration provides co-located measurements within the tropopause region, minimizing calibration offsets and temporal mismatches. A vertical separation of 100–150 m allows instantaneous determination of gradients and mixing signatures that would otherwise be obscured by sequential profiling. Compared with previous approaches, TPC-TOSS thus offers a unique capability to quantify small-scale transport and mixing processes at the tropopause and to relate observed gradients to underlying dynamical mechanisms.

In the following sections we will present the new setup and will provide uncertainties and individual tests, as well as some examples demonstrating the agreement between the two platforms. Additionally, we will showcase typical results achieved during the first field setup.

## 2 The TPEx I Intensive Operation Period in June 2024

Figure 1. (a) Overview over all conducted measurement flights during the TPEx I mission in June 2024. The red coloured flight paths are flights with the TPC-TOSS deployed whereas the black lines show the other flights without TPC-TOSS. The map was created from public-domain GIS data found on the Natural Earth website (http://www.naturalearthdata.com, last access: 30 June 2025). (b) Schematic of the concept of the dual platform approach with TPC-TOSS attached to the Learjet aircraft with a steel wire rope allowing for simultaneous measurements at two levels. Colours in the background represent an arbitrary air mass property changing from low to high values at the tropopause. This property can be measured simultaneously by the two platforms. Modified from Emig et al. (2025).

The aircraft campaign TPEx I (TropoPause composition gradients and mixing Experiment) is the central aircraft mission in the collaborative research center TPChange (The TropoPause region in a Changing atmosphere) and took place between 10 and 21 June 2024 based at Hohn airfield in Northern Germany (54°18′49″N, 9°32′17″E). The TPEx I mission addressed questions regarding the water vapor distribution in the upper troposphere and lower stratosphere (UTLS), the identification of mixing at and across the extratropical tropopause induced by diabatic processes and the source apportionment of aerosols and icenucleating particles (INPs) for understanding the main pathways of transport into the UTLS. Additionally, the mission aimed for studying vertical transport of aerosol particles and trace gases from the planetary boundary layer (PBL) into the UTLS and the effect on the chemical composition of the UTLS and new particle formation (NPF) events as well as cloud particle and cirrus formation.

Therefore, we equipped the research aircraft, a Learjet 35A, with a set of in-situ measurements of trace gases (e.g., CO and ozone) and aerosol quantities (particle size distribution and chemical composition) as well as offline samplers of aerosol and cloud particles and INPs. Furthermore, we used the unique approach of a fully automated towed sensor shuttle (TPC-TOSS) attached to the aircraft with partly redundant instrumentation and deployed it during four out of eight scientific flights during TPEx I (Table 1 and Fig. 1a and b). The TPC-TOSS was attached to the aircraft via a purely mechanical connection using a steel wire rope. This approach provides observational data of vertical gradients of quantities, such as potential temperature,

Table 1. Overview of the research flights with TPC-TOSS during TPEx I.

| Flight Nr. | Date         | Time              | Region     |
|------------|--------------|-------------------|------------|
| F03        | 11 June 2024 | 10:30 - 13:07 UTC | Baltic Sea |
| F06        | 14 June 2024 | 07:07 - 09:18 UTC | Baltic Sea |
| F07        | 17 June 2024 | 07:01 - 09:35 UTC | North Sea  |
| F10        | 20 June 2024 | 07:06 - 09:47 UTC | North Sea  |
|            |              |                   |            |

ozone mixing ratios and particle size distribution for aerosol particles between 95 nm and 1  $\mu$ m. The vertical distance between both platforms is around 150 m in tropopause regions and therefore aimed at resolving transient small scale variability of the tropopause structure and composition induced by small-scale processes (e.g., strong shear or small scale processes within extended cirrus decks).

As a consequence of safety constraints flights with the TPC-TOSS were only allowed in restricted air spaces. For TPEx I we used restricted air spaces over the Baltic Sea close to Usedom and over the North Sea west of Helgoland for the measurement flights. During TPEx I the Learjet reached maximum altitudes of 12000 m without the towed sensor shuttle and 10900 m with TPC-TOSS deployed. The maximum flight time was around 2.5 h with and up to 4 h without TPC-TOSS. During the mission in total eight research flights and one test flight were conducted of which four flights used the dual platform approach, two in each restricted air space (Fig. 1a).

The scientific flight planning during the mission was performed with the help of high resolution model forecast from the ICON-D2 (ICOsahedral Nonhydrostatic) and the ECMWF (European Centre for Medium-Range Weather Forecasts) model with additional output of the CLaMS-Ice (Chemical Lagrangian Model of the Stratosphere) model for cirrus cloud predictions. For operational planning of the flights we used the Mission Support System (MSS, Bauer et al. (2022)) with meteorological and chemical data from ECMWF from the IFS and CAMS forecast models. MSS as a server client application allows to interactively plan flight trajectories based on current four dimensional forecast data. Additionally, we used high resolution data from ICON-D2 for forecasts of convection as well as from ICON for WCB forecasts.

The meteorological conditions in June 2024 were quite favorable for the scientific objectives of the mission. In this period several low pressure systems crossed the measurement region. The outflow of associated warm conveyor belts (WCBs) was within the range of the Learjet 35A and could be studied (Joppe et al., 2025). Furthermore, highly variable tropopause heights and convection over Germany and parts of Sweden (Konjari et al., 2025) were probed.

### **Instrumentation during TPEx I**

For the TPEx mission the measurement platforms (Learjet including underwing pod "Knuffi" and TPC-TOSS) were equipped with instrumentation for in situ trace gas measurements, INP characterisation, aerosol number concentration, size distribu-

**Table 2.** Overview of the instrumentation of the **Learjet** as well as the measured quantities.

| Instrument               | Measured Quantities, Range                          | Sampling<br>Frequency | Uncertainty                           | Reference                 |
|--------------------------|-----------------------------------------------------|-----------------------|---------------------------------------|---------------------------|
| UMAQS <sup>1</sup>       | CO, N <sub>2</sub> O, 1–5000 ppbv,                  | 1 Hz                  | 0.6 ppbv (CO),                        | Müller et al. (2015)      |
|                          |                                                     |                       | $0.18~\mathrm{ppbv}~(\mathrm{N_2O})$  | Kunkel et al. (2019)      |
| FISH <sup>2</sup>        | Gas phase water vapor, 1-1000 ppmv                  | 1 Hz                  | $7~\% \pm 0.3~\mathrm{ppmv}$          | Rolf et al. (2024)        |
|                          |                                                     |                       |                                       | Meyer et al. (2015)       |
|                          |                                                     |                       |                                       | Zöger et al. (1999)       |
| WaSul <sup>3</sup>       | Total water vapor, 0.5–60000 ppmv                   | $0.5~\mathrm{Hz}$     | 21 %                                  | Tátrai et al. (2015)      |
|                          |                                                     |                       |                                       | Rolf et al. (2024)        |
| FRIDGE/SEM <sup>4</sup>  | INP concentration and physico-chemical              | aerosol               | 22 % (INP                             | Schrod et al. (2016)      |
|                          | properties                                          | sampling              | concentration)                        | Schneider et al. (2024)   |
|                          | (elemental composition $> 80 \text{ nm}$            | $1560 \min$           |                                       |                           |
|                          | size and morphology > 20 nm)                        |                       |                                       |                           |
| CARIBIC-AMS <sup>5</sup> | Chemical composition of non-refractory              | $30 \mathrm{\ s}$     | 30% (total mass                       | Schneider et al. (2025)   |
|                          | aerosol particles, $50-800 \mathrm{\ nm}$           |                       | concentration)                        | Bahreini et al. (2009)    |
|                          |                                                     |                       |                                       | Middlebrook et al. (2012) |
| SkyOPC <sup>6</sup>      | Particle size distribution, 250 nm–3 $\mu m$        | 6 s                   | 6% (total number                      | Bundke et al. (2015)      |
|                          |                                                     |                       | concentration)                        |                           |
| UHSAS-C <sup>7</sup>     | Particle size distribution, $0.095-1~\mu\mathrm{m}$ | 1 Hz                  | 1030~% (total number                  | Cai et al. (2008)         |
|                          |                                                     |                       | concentration)                        |                           |
| MC-CPC <sup>8</sup>      | Aerosol number concentration from $12~\mathrm{nm}$  | 1 Hz                  | 22 %                                  | Richter et al. (2025)     |
|                          | and 16 nm up to µm range                            | 1 Hz                  |                                       |                           |
| ICH9-sensor              | Temperature and relative humidity                   | 1 Hz                  | $0.32 \text{ K (T)}, 5 \% (RH_{liq})$ | Helten et al. (1998)      |
| GNSS/INS <sup>10</sup>   | Position, attitude and velocity                     | 1 Hz                  | 1.25 m (hor. pos.),                   |                           |
|                          |                                                     |                       | 2 m (ver. pos.),                      |                           |
|                          |                                                     |                       | $0.05^{\circ}$ (roll, pitch),         |                           |
|                          |                                                     |                       | $0.25^{\circ}$ (heading),             |                           |
|                          | Acceleration                                        | $100~\mathrm{Hz}$     | $0.05~\mathrm{ms}^{-1}$               |                           |

<sup>&</sup>lt;sup>1</sup> University of Mainz Airborne Quantum Cascade Laser Spectrometer. <sup>2</sup> Fast In-situ Stratospheric Hygrometer. <sup>3</sup> Water vapor / Sulfur dioxide monitor. <sup>4</sup> FRankfurt Ice nucleation Deposition freezinG Experiment/Scanning Electron Microscope. <sup>5</sup> Civil Aircraft for the Regular Investigation of the Atmosphere Based on an Instrument Container – Aerosol Mass Spectrometer. <sup>6</sup> Sky Optical Particle Counter. <sup>7</sup> Ultra-High Sensitivity Aerosol Spectrometer. <sup>8</sup> MultiChannel – Condensation Particle Counter. <sup>9</sup> IAGOS (In-service Aircraft for a Global Observing System) capacitive hygrometer. <sup>10</sup> Global Navigation Satellite System / Inertial Navigation System.

tion and composition measurements based on online (in situ) and offline filter analysis methods. These measurements were supplemented by measurements for meteorological parameters (temperature, humidity and pressure) as well as flight altitude, positioning and acceleration information using a GNSS/INS (Global Navigation Satellite System / Inertial Navigation System)

Table 3. Overview of the instrumentation of the Underwing pod "Knuffi" as well as the measured quantities.

| Instrument                                                                       | Measured Quantities, Range                      | Sampling<br>Frequency | Uncertainty                      | Reference                |  |
|----------------------------------------------------------------------------------|-------------------------------------------------|-----------------------|----------------------------------|--------------------------|--|
| 2BTech Ozone                                                                     | $O_3$ , 0–20 ppmv                               | 0.5 Hz                | $\pm  (3 \text{ ppbv} + 0.7 \%)$ | Johnson et al. (2014)    |  |
| NIXE-CAPS <sup>1</sup>                                                           | Number concentration and size distribution      |                       | 20~%                             | Meyer (2013)             |  |
|                                                                                  | of cloud particles, 0.61–937 $\mu m$ (diameter) | 1 Hz                  |                                  | Krämer et al. (2020)     |  |
| $SOAP^2$                                                                         | Organic aerosol molecular composition,          |                       | 1020~% (low-                     | Breuninger et al. (2025) |  |
|                                                                                  | 10–2000 nm,                                     |                       | volatile organics)               |                          |  |
|                                                                                  | five samples including one filter blank         | 15-140 min each       |                                  |                          |  |
|                                                                                  | per flight                                      |                       |                                  |                          |  |
| $BCP^3$                                                                          | concentration of particles with an optical      |                       | 21 %                             | Beswick et al. (2014)    |  |
|                                                                                  | equivalent diameter between 5-75 μm,            | 1 Hz                  |                                  |                          |  |
|                                                                                  | $0.002 - 20 \text{ cm}^{-3}$                    |                       |                                  |                          |  |
| Offline particle composition, particle size and particle shape, aerosol sampling |                                                 |                       |                                  |                          |  |
| MultiMINI8 <sup>4</sup>                                                          | 30 nm–10 μm                                     | sampling 1–30 min     |                                  | Ebert et al. (2016)      |  |
| SPAFiS <sup>5</sup>                                                              | $100 \ \mathrm{nm}$ – $10 \ \mu\mathrm{m}$      | sampling 1–30 min     |                                  |                          |  |
| NanoPS <sup>6</sup>                                                              | $\leq 500 \; \mathrm{nm}$                       | sampling 1–30 min     |                                  |                          |  |

<sup>&</sup>lt;sup>1</sup> New Ice eXpEriment - Cloud and Aerosol Particle Spectrometer. <sup>2</sup> Sampler for Organic Aerosol Particles. <sup>3</sup> Back-Scatter Cloud Probe. <sup>4</sup> Multi stage Micro INertial Impactor sampler 8. <sup>5</sup> Single PArticle Filter Sampler. <sup>6</sup> Nano Particle Sampler.

125

navigational sensor. In particular the relative position between Learjet and TPC-TOSS is of major importance for the measurements with the dual platform approach. Furthermore, NIXE-CAPS (New Ice eXpEriment - Cloud and Aerosol Particle Spectrometer), installed in an underwing pod attached to the left wing of the Learjet, allows for the characterisation of cloud particles (number concentration, size). Some of the measurements were performed simultaneously on Learjet and TPC-TOSS (Table 2 to Table 4). This synchronized payload with partly identical instrumentation on TPC-TOSS and Learjet in particular allows for the determination of gradients of different quantities which in turn are used to study the effect of small-scale transient features on the UTLS composition. Therefore, the focus of this paper is on the TPC-TOSS and the duplicated instrumentation on the Learjet as part of the dual platform approach. For all other instrumentation on the Learjet and in the underwing pod we provide references for characterization and application of the respective instrument in Table 2 and Table 3. In Sect. 4 we will describe the TPC-TOSS instrumentation in detail also demonstrating the comparability and similar performance of the instruments based on cross calibration in the laboratory, during the intensive operation period on ground and also during research flights. This is in particular essential to ensure that observed differences between the two payloads reflect true atmospheric gradients rather than instrumental offsets.

**Table 4.** Overview of the instrumentation of the **TPC-TOSS** as well as the measured quantities.

| Instrument               | Measured Quantities, Range                                                    | Sampling<br>Frequency | Uncertainty                                     | Reference             |
|--------------------------|-------------------------------------------------------------------------------|-----------------------|-------------------------------------------------|-----------------------|
| 2BTech Ozone             | O <sub>3</sub> , 0–20 ppmv                                                    | 0.5 Hz                | $\pm (3 \text{ ppbv} + 0.9 \%)$                 | Johnson et al. (2014) |
| UHSAS-A <sup>1</sup>     | Particle size distribution, 0.095–1 $\mu \mathrm{m}$                          | 1 Hz                  | 10–30 % (total number concentration)            | Mahnke et al. (2021)  |
| $BCP^2$                  | concentration of particles with an optical                                    |                       | 21 %                                            | Beswick et al. (2014) |
|                          | equivalent diameter between 5–75 $\mu\mathrm{m},$ $0.002–20~\mathrm{cm}^{-3}$ | 1 Hz                  |                                                 |                       |
| ICH <sup>3</sup> -sensor | Temperature and relative humidity                                             | 1 Hz                  | $0.32~{\rm K}$ (T), $5~\%$ (RH <sub>liq</sub> ) | Helten et al. (1998)  |
| GNSS/INS <sup>4</sup>    | Position, attitude and velocity                                               | 1 Hz                  | 1.25 m (hor. pos.), 2 m (ver. pos.)             |                       |
|                          |                                                                               |                       | $0.05^{\circ}$ (roll, pitch)                    |                       |
|                          |                                                                               |                       | $0.25^{\circ}$ (heading)                        |                       |
|                          | Acceleration                                                                  | $100~\mathrm{Hz}$     | $0.05~{\rm ms}^{-1}$                            |                       |

<sup>&</sup>lt;sup>1</sup> Ultra-High Sensitivity Aerosol Spectrometer. <sup>2</sup> Back-Scatter Cloud Probe. <sup>3</sup> IAGOS (In-service Aircraft for a Global Observing System) capacitive hygrometer. <sup>4</sup> Global Navigation Satellite System / Inertial Navigation System.

# 3 Technical design TPC-TOSS

135

The technical design of TPC-TOSS builds on an earlier version of the towed sensor shuttle (AIRTOSS) described in Klingebiel et al. (2017, and references therein). For TPEx I in June 2024, modifications were necessary to adapt the TPC-TOSS for trace gas and aerosol measurements. The body of the TPC-TOSS has a length of 2.57 m, a diameter of 0.24 m and a net weight of 27 kg. It is capable of carrying a maximum payload of 43 kg. The TPC-TOSS payload during TPEx I is summarized in Table 4 and described in more detail in Sect. 4. Individual instruments and additional equipment can be mounted on an internal aluminum frame that is split in three sections (Fig. 2b). The front section mainly contains the Ultra-High Sensitivity Aerosol Spectrometer (UHSAS-A; "A" denotes the airborne version within TPC-TOSS, "C" in Table 2 denotes the Learjet cabin version) (Sect. 5.3). In the middle part of the TPC-TOSS the battery pack is mounted, which contains 8 lithium iron phosphate accumulators controlled by a battery management system. Individual cells are configured as such that an output voltage of 25.6 VDC is provided to supply all instrumentation within the TPC-TOSS as there is only a mechanical connection with a steel wire rope between aircraft and TPC-TOSS. The capacity for the battery pack is 50 Ah which allows for the operation of the TPC-TOSS instrumentation for 6-7 h, which exceeds the maximum length of a research flight determined by fuel and Learjet performance. The rear section of the TPC-TOSS internal structure contains the Back-Scatter Cloud Probe (BCP, directly attached to the drag body cover), the 2BTech ozone monitor model 205, GNSS/INS instrumentation and data acquisition. The IAGOS (In-service Aircraft for a Global Observing System) (Petzold et al., 2015) capacitive hygrometer (ICH) for temperature and humidity measurements (rear part), the ozone bypass inlet and outlet (rear part) as well as the two GPS

**Figure 2.** (a) TPC-TOSS with ozone bypass inlet and outlet. The ozone instrument is located in the back of the TPC-TOSS. Note that, while attached to the aircraft, the TPC-TOSS is rotated upward by 90 °. (b) Computer-Aided Design (CAD) drawing of the TPC-TOSS including the instrumentation and battery pack.

antenna (rear and front part) are mounted directly on the drag body cover which is made of glass-fibre reinforced plastic. As mentioned before, there were some modifications of the body cover necessary to mount the GPS antenna, BCP and the trace gas inlets. The mounting plates for these components were manufactured based on 3D printing to reduce weight.

The ozone instrument was connected to a bypass type tubing system consisting of 1/4" teflon tubing. While the bypass inlet was mounted forward facing, the bypass outlet was mounted backward facing allowing for a high bypass flow of 20–30 slpm to reduce the residence time inside the tubing (Fig. 2). The ozone instrument sampled from the main bypass inlet line using a T-type insertion. The forward facing stainless steel inlet for the UHSAS-A aerosol sampling was part of the main instrument. An internal pump actively maintained sample and sheath flow of 50 and 700 cm<sup>3</sup> min<sup>-1</sup>, respectively.

155

160

The total weight and power consumption of all instrumentation within TPC-TOSS amounts to 41 kg and 185 W. The individual components within the TPC-TOSS frame and attachments to the towed sensor shuttle cover are positioned to locate the center of gravity of the TPC-TOSS close to the hook (distance of only 120 mm), where the steel wire connecting the TPC-TOSS to the Learjet is attached. The position of the center of gravity is crucial for a stable horizontal position during flight. Air brakes on the wings of the TPC-TOSS further support maintaining a stable flight attitude.

The TPC-TOSS is attached to a winch under the right aircraft wing that is equipped with a steel wire of a maximum length up to 4 km. The pilots operate the winch to release the drag body to the desired wire length and retract it after the measurements.

For certification reasons the operation of the winch is only allowed below 25000 ft (7.6 km) while the maximum flight altitude with the TPC-TOSS deployed is 41000 ft (12.5 km). During the TPEX I flights with the TPC-TOSS a wire length of 3000 ft (914 m) was used. The main reason for not using a longer wire length was the military controlled restricted air space with a maximum side length of 50–80 km in which we were only allowed to fly with TPC-TOSS due to safety constraints. The small area resulted in multiple turns during aircraft operation. Based on the experience from earlier campaigns in the same airspace, the chosen wire length was a compromise between a maximum reachable vertical distance between Learjet and TPC-TOSS and safe and feasible Learjet operation (Klingebiel et al., 2017, and references therein). With this wire length a vertical distance between Learjet and TPC-TOSS of 152 ± 8 m was reached during stable flight conditions (no turns or climbs/descents). The maximum range of vertical distance was between 95 m and 220 m including turns and altitude changes. Further details on the relative position of TPC-TOSS and Leariet are discussed in Sect. 5.1.

## 4 Instrument characterization

The following section describes and characterizes the redundant instrumentation installed on both platforms (Learjet including underwing pod and TPC-TOSS) which are a crucial part of the novel dual platform approach to measure ozone and aerosol gradients in the UTLS region.

## 4.1 GNSS/INS

175

To analyze co-located measurements on two platforms with respect to gradients of aerosol, trace species and meteorological parameters one requires a precise definition of the position of the individual platform and/or the relative position between both, Learjet and TPC-TOSS. We used a high performance tactical grade GNSS-Aided Inertial Navigation System (GNSS/INS) which uses MEMS (Micro-Electro-Mechanical Systems) inertial sensors in combination with dual multi-frequency GNSS receivers. The used 3DM-GQ7 (MicroStrain company) consists of a 3-axis accelerometer, 3-axis gyroscope, 3-axis magnetometer, a pressure altimeter and a dual GNSS receiver. The system performance given as uncertainty (1 σ) of the most important parameters within the operating temperature range of –40–85 °C is based on manufacturer information as follows: The uncertainty of horizontal and vertical position amounts to 1.25 m and 2 m, respectively. With respect to flight attitude, roll and pitch angles could be derived with an uncertainty of 0.05° while the error for the heading amounts to 0.25°. The error of the measured velocity is 0.05 ms<sup>-1</sup>.

This sensor was installed on both the Learjet and the TPC-TOSS to get consistent information on position and attitude of the respective platform. Two GPS antennas were installed on each platform located at a horizontal distance of 198 cm on the Learjet and 148 cm on the TPC-TOSS. The use of two antennas (L1 band at 1600 MHz and L2 band at 1200 MHz) increases the redundancy of the GPS positioning in case one antenna experiences reception issues. Simultaneously, it enables improved heading determination based on the relative position of the two antennas, with the associated uncertainty stated above. Position and attitude information were recorded at a resolution of 1 Hz. In addition, acceleration data were available at 100 Hz, providing insight into turbulent flight conditions experienced by the Learjet and TPC-TOSS.

## 4.2 IAGOS Capacitive hygrometer ICH

Relative humidity with respect to liquid phase water (RH<sub>liq</sub>) was measured using an instrument that is also employed in the IAGOS program. The IAGOS capacitive hygrometer (ICH) was mounted in the TPC-TOSS. The ICH, which also measures temperature, consists of a thin-film HUMICAP® capacitive sensor (Vaisala, Finland) whose capacitance depends on the relative humidity of the dielectric layer of the condenser, and a platinum resistance sensor (Pt100) that measures the temperature at the humidity sensing surface. The sensor itself and the applied calibration techniques are described in detail by Helten et al. (1998). The measurement principle is based on the diffusion-limited adsorption of the H<sub>2</sub>O molecules by the dielectric membrane of the sensor. Since diffusion is strongly temperature-dependent, the sensor's response slows down from seconds to a few minutes with decreasing temperatures. The relative humidity and temperature signals are fed into a microprocessor-controlled transmitter unit (HMP230, Vaisala) which passes the signals to the data acquisition system. The data conversion from capacitance signals to relative humidity values is performed offline in a separate data quality assurance and analysis step (Neis et al., 2015a, b). The ICH sensor is mounted at the top of an axisymmetric sensor carrier, which is designed for installation in an appropriate housing (ICH-RS: Model 102 BX, Rosemount Inc., Aerospace Division, USA). The ICH sensor is designed for routine autonomous measurements aboard passenger aircraft. Its passive measurement technique requires no sampling line and pump, thus low demand of maintenance. Before the installation on the aircraft and after 500 h flight hours ( $\sim$  four to six weeks) within the IAGOS framework, an individual calibration of each ICH sensor is necessary, which is accomplished in the environmental simulation chamber at Jülich (Smit et al., 2000). During the TPEx I campaign, the ICH sensor was calibrated before and after the campaign. These calibrations are made over a sensor temperature range between -40-20 °C against a frost point hygrometer (MBW373) at 2–50 % RH $_{liq}$  with a temperature accuracy of  $\pm 0.1$  K. During flight, in fact, the ambient air is adiabatically compressed in the housing, leading to a significant temperature increase of the air sampled by the sensor (up to 30 °C). Therefore, -40 °C sensor temperature, namely the lowest temperature of the calibration, corresponds to -70 °C in the real atmosphere, which is rarely reached at aircraft cruising altitude. The adiabatic heating is corrected using the Mach number after true aircraft speed (Neis et al., 2015a).

Based on chamber calibration with the MBW373 frost-point mirror and intercomparison with airborne instruments in research aircraft measurement missions (Neis et al., 2015a, b; Rolf et al., 2024), the temperature and  $RH_{liq}$  uncertainties produced by the ICH sensor are  $\pm 0.32$  K and  $\pm$  (5–6) %, respectively. Therefore, relative humidity with respect to ice ( $RH_{ice}$ ) and water vapour mixing ratio can also be provided calculated from  $RH_{liq}$  and temperature.

### 225 4.3 Back-Scatter Cloud Probe (BCP)

The BCP is part of the IAGOS system, and is a compact, lightweight, near-field and single particle backscattering optical spectrometer to measure the concentration and optical equivalent diameter of particles from 5 to 75  $\mu$ m (Beswick et al., 2014). The BCP features a laser diode emitting focused and linearly polarized light at 658 nm, which passes through a heated glass window in the aircraft skin and focuses on a small region approximately 4 cm away. Light scattered back at a solid angle of

144–156° by particles in the sample volume is collected by lenses and focused onto an avalanche photodiode for detection. 230 The cloud particle number concentration is calculated from the sampling area times the true air speed of the aircraft. It was primarily designed as a real-time qualitative cloud indicator for data quality control of trace gas instruments of the IAGOS system. Subsequent evaluations and investigations (Petzold et al., 2017; Lloyd et al., 2021) reveal that the BCP cloud dataset is also of use for the study of contrail and natural cirrus. Limited by the detectable particle size range, BCP is insufficient for rather small (< 5 µm) and large cirrus particles up to the size of approx. 1 mm in cirrus clouds. The total measurement concentration 235 ranges from 0.002 cm<sup>-3</sup> to approx, 20 cm<sup>-3</sup> in cirrus clouds, as observed during IAGOS cruising condition. High cloud particle number concentrations up to 200 cm<sup>-3</sup> in liquid water clouds were demonstrated to be within the detectability of the BCP by Beswick et al. (2014). Assuming the sample area as reported by Beswick et al. (2014) and a typical mean aircraft cruising speed of 250 ms<sup>-1</sup>, the estimated lower threshold for cloud particle detection with a temporal resolution of 4 s (IAGOS 240 operation conditions) would be 0.015 cm<sup>-3</sup>, but with a sampling uncertainty of 50 % according to Poisson statistics (Petzold et al., 2017). During TPEx I, the BCP was integrated into TPC-TOSS and the underwing pod "Knuffi" as it is installed at the fuselage during IAGOS routine measurements.

### 4.4 Ozone measurements

During TPEx I, ozone was measured using two modified 2BTech model 205 instruments (Johnson et al., 2014). The measurements are based on the absorption of UV at the wavelength 254 nm at ambient pressure. Small pumps are used to continuously purge the instruments at a flow rate of 1.7 lmin<sup>-1</sup> for the underwing pod instrument and 2.6 lmin<sup>-1</sup> for the TPC-TOSS instrument at ground pressure. The flow difference arises from the change from a two-pump to an only one-pump flow scheme for newer versions of the instrument. The instrument was modified for operating at high altitudes within the underwing pod and the TPC-TOSS under low pressure and low temperature conditions. These modifications consisted of an upgrade of the pressure sensor suitable and calibrated for an altitude range up to 25 km, the pump, an additional lamp heater to improve the stability of the UV source, and the addition of of insulation to protect the instrument by maintaining temperatures above 0 °C. The instrument is equipped with two absorption cells of which one (cell A) is purged with ambient air at a 2 s time interval to determine the light intensity  $I(t_0)$ . The air stream for the second cell (cell B) in this time interval is led through an Hopcalite ozone scrubber to remove any ozone to determine the light intensity  $I_0(t_0)$  without any absorber present in the cell. During the subsequent 2 s time interval  $[t_0, t_1]$  cell A is purged with ozone scrubbed air while cell B is purged with ambient air. At time  $t_1$  an ozone value is calculated for cell A applying Beer's law with the measured ratio of absorption signals  $I(t_0)$  and  $I_0(t_1)$ . For cell B an ozone value is calculated using  $I_0(t_0)$  and  $I(t_1)$ . The ozone value for each cell is further converted into a mixing ratio by applying the measured temperature in the respective cell and the cell pressure and finally stored as the average of both cells at  $t_1$ . At time  $t_2$  the ozone value for cell A is calculated using again  $I_0(t_1)$  and the new absorption signal  $I(t_2)$  while for cell B ozone is determined from  $I(t_1)$  and  $I_0(t_2)$ . Again, the average ozone mixing ratio based on both cells is stored at  $t_2$ . As a consequence every individual value in each measurement cell is used twice for the calculation of subsequent mixing ratios leading to an fully independent determination of an ozone value only every 4 s. To account for drifts and asymmetries in the measurement cells, the streams of ambient air and scrubbed air through the cells are switched every 2 s. The minimum time resolution therefore is 2 s, corresponding to approximately 300 m spatial resolution during flight operation.

During TPEx I one instrument was mounted outside the pressurized cabin into the underwing pod ("Knuffi") at ambient pressure. The other instrument was placed into the TPC-TOSS. Temperature and pressure dependencies were characterized in the laboratory before the deployment during TPEx I (Sect. 4.4.3, 4.4.4). To avoid operational temperatures dropping to values below the instrument specifications, both devices were thermally isolated. Due to safety reasons, the TPC-TOSS had to be powered off while being attached to the aircraft, therefore no active heating could be applied until TPC-TOSS was released.

#### 4.4.1 Noise and drift

**Figure 3.** Allan-Werle-plots for the ozone instruments installed on the Learjet (a) and TPC-TOSS (b). Upper panels show the ozone time series from ground tests during the field campaign (8 June 2024 for TPC-TOSS ozone and 9 June 2024 for "Knuffi" ozone) used to calculate the Allan variance.

Both ozone instruments were extensively checked and tested prior and during the campaign. Noise and drift (as a measure of stability) under different laboratory conditions and during the field campaign on ground have been checked.

To test noise and drift of both instruments we used the Allan variance (Allan, 1966; Werle, 2011). Having pure statistical noise (white noise) the Allan variance should decrease with increasing integration time following the black solid line in Fig. 3 (lower panel). Our instruments show both a constant or even increasing Allan variance up to integration times of 4–6 s followed by a decrease of the Allan variance until an optimal integration time of 300–500 s where the Allan variance is at minimum. At larger integration times the slow drift starts to dominate leading to increasing Allan variance. Furthermore a significant deviation from the white noise floor (black line) is observed. According to Werle (2011), additional non-white noise components, e.g., flicker noise, could lead to this deviation. A maximum of the Allan variance at an integration time larger than the lowest time resolution (in our case 2 s) corresponds to a low pass filter characteristic. The low pass filter is usually applied in the frequency domain but its effect could also be observed in the time domain. Signal smoothing or damping effects

of fast concentration changes would in turn lead to the observed behavior of the Allan variance. As discussed in Section 4.4, the output frequency of the ozone instrument is 2 s but the measurement process itself leads to independent data points only every 4 s. Taking into account that the gas exchange time in the tubing system is on the order of 1 s, data points before the Allan variance maximum are to some degree correlated. This is similar to applying a smoothing or running mean of 4–6 s to the ozone data which increases the Allan variance with integration time. After the maximum, the Allan variance further decreases with integration time but deviates from the black solid line. For the "Knuffi" instrument a significant decrease is observed only after 30 s. For both instruments we observed slow irregular changes of the cell pressure caused by irregular changes of pump capacity of the small internal membrane pumps. These variations most probably add non statistical noise components (flicker noise) to the Allan Variance expressed as the observed deviation from the black line (Werle, 2011). As these cell pressure variations are stronger for the "Knuffi" instrument, the Allan variance for this instrument is stronger affected during the first 20 to 30 s compared to the TPC-TOSS instrument. Similar results for the Allan variance with an Allan maximum around 4 s are reported by Moormann et al. (2025), who operated the same type of instrument on a drone, thereby confirming our laboratory and field tests.

Based on these Allan variance analysis the noise of both instruments under laboratory conditions amounts to 1 ppbv (1  $\sigma$ ) for 2 s data at a mixing ratio of 200 ppbv. The quantification of the drift of both instruments as a measure of stability is done in the following section.

## 4.4.2 Linearity

To test the linearity of the ozone monitor model 205 we checked the instruments against a calibration source (2B Tech ozone calibration source model 306). The calibration source is calibrated against a NIST (National Institute of Standards and Technology) traceable standard and is capable of producing ozone with an accuracy and precision better than 1 ppbv in the range 30–100 ppbv ozone or 1 % in the range 95–1000 ppbv (Birks et al., 2018).

We tested both instruments in their final mechanical configuration in the field to account for the effect of different inlet lengths. The instruments were both connected to the calibration source and purged with calibrated ozone for ten minutes for each mixing ratio. The mixing ratio was stepwise increased from 100 to 900 ppbv. Figure 4 shows that both instruments exhibit a linear response over the expected data range, deviating from unity by 3.6 % and 1.6 %, respectively (relative to the factory settings for the implemented gain and offset parameters). The offset was zero within the statistical uncertainty. It is important to note that the newly derived offset and gain parameters were implemented as calibration parameters during post-processing of the ozone data. In this step, the measured ozone values were corrected by applying the offset and gain to the raw data.

We repeated the calibration procedure between research flights and after the campaign, which allows for assessing stability by analyzing the drift in offset and gain during the regular calibrations. For the TPC-TOSS ozone instrument, the offset and gain drifted by 2.2 ppbv and 0.7 % (1  $\sigma$ ), respectively. For the "Knuffi" instrument, the stability parameter were 2.2 ppbv for the offset and 0.1 % (1  $\sigma$ ) for the gain.

Based on these regular calibrations, we further analyzed the instrument noise over a broader range of mixing ratios, covering the measured mixing ratios during research flights, and compared the results to the Allan variance presented in the previous

**Figure 4.** Linearity check of the ozone instruments using the 2B Tech ozone calibration source model 306 in between the research flights on 19 June 2024. Each mixing ratio step consists of three minutes of data after ozone has stabilized at the respective target mixing ratio. (a) results for the Learjet instrument inside "Knuffi" and (b) for the TPC-TOSS instrument.

section. This analysis indicates that a mixing ratio dependent noise component needs to be added. For both the TPC-TOSS and the "Knuffi" instrument the noise for the final mechanical setup is 2 ppbv + 0.5 %. Applying Gaussian error propagation the total uncertainty, which includes both noise and stability, is 3 ppbv + 0.9 % for the TPC-TOSS and 3 ppbv + 0.7 % for the Learjet instrument within the underwing pod.

#### 4.4.3 Temperature dependence

The two ozone monitors model 205 were mounted outside the cabin and thus affected by cold ambient air temperatures during flight. Both instruments were equipped with a cold weather upgrade including lamp heating and a pump capable of operating at temperatures below freezing point. In addition, both instruments were thermally isolated using Basotect® foam to prevent them from cooling below specified temperature ranges during operation. As mentioned before, the ozone instrument on the TPC-TOSS could only be switched on after the TPC-TOSS was released from the Learjet, which was typically under cold conditions.

To identify temperature dependencies of the measured ozone data, both instruments were tested in a cold chamber being capable of operating at -20 °C. In the following we only show results from the TPC-TOSS instrument as both instrument show the same behavior inside the cold chamber. The ozone instrument was placed inside the cold chamber and purged with calibration gas at 50 ppbv from the ozone calibration source model 306. The cold chamber was initially operated at approximately -5 °C and was cooled down to -20 °C once the ozone monitor was running inside.

Figure 5. Temperature dependence test of the ozone monitor within a cold chamber. (a) ozone mixing ratio is shown in red. The cell temperature within the ozone monitor is shown in pink and further internal temperature measurements are shown in green colours. Two sensors measuring the air temperatures at different positions inside the cold chamber are shown in blue colours. The dark blue measurement is nearby the ozone monitor and the light blue sensor was located below the roof of the cold chamber. The black line in the upper panel shows the target ozone mixing ratio of 50 ppbv, the grey line in the lower panel shows the target temperature of -20 °C of the cold chamber. (b) histogram of ozone data during the temperature test. Red dots show the distribution of ozone measurements. The black line shows a gaussian curve fit to the measured ozone distribution.

Figure 5a shows the evolution of ozone mixing ratios and internal temperatures of the ozone instrument. Temperature "air" is measured in the air inside the box of the ozone monitor, temperature "lamp" is measured on the lamp housing and temperature "plate" is measured on the plate where all electronics and optics are mounted on. At constant cold chamber temperature, the ozone cell temperature and additional temperatures inside the ozone instrument were initially stable. They started decreasing as soon as the cold chamber was cooled down to approximately –20 °C. After 75 min of operation at –20 °C, the ozone related temperatures began to stabilize at a level approximately 10 °C lower, but still well above freezing point.

Ozone mixing ratios were quite constant during this experiment showing only a very weak drift of  $1.28~\mathrm{ppbv}$  during two hours of measurements. This drift is within the uncertainty range of ozone generation (1 ppbv in the range 30– $100~\mathrm{ppbv}$  ozone) and measurement (3 ppbv + 0.9~%). The histogram (Fig. 5b) confirms a statistical distribution of the values and thus no indication for non-linearities. The standard deviation of the gaussian fit amounts to  $1.6~\mathrm{ppbv}$  and is on the order of the instrument noise determined in the previous section. Based on these cold chamber tests no temperature dependencies of the ozone monitors was expected.

As outside air temperatures during the research flights were lower than those in the cold chamber tests, we further analyzed the temperature behavior of the ozone instruments during research flight F10 on 20 June 2024 in more detail. Figure 6 shows that cell temperatures within the ozone instruments stayed well above freezing point during research flight F10 despite outside air

**Figure 6.** Upper panel: Ozone mixing ratios measured with the ozone monitors in the "Knuffi" (blue) on the Learjet and in the TPC-TOSS (light brown). Middle panel: Temperature evolution of measured cell temperatures for the TPC-TOSS (orange) and "Knuffi" instrument (brown). Green temperature curves represent temperatures measured inside the housing of the ozone monitor. Lower panel: Black and grey dots represent GPS altitude of the Learjet and TPC-TOSS and dark and light red dots represent outside air temperatures measured on Learjet and TPC-TOSS by the ICH, respectively.

temperatures dropped below -50 °C at highest flight levels. The temperature measured on the lamp housing closely follows the cell temperature, the three other temperature measures reach values below zero during the second half of the flight. These temperatures are measured in the air inside the box of the ozone monitor (temperature "air"), on the detector housing (temperature "detector") and on the plate where all electronics and optics are mounted on (temperature "plate"). This configuration is similar to the cold chamber test before the campaign. Temperatures below zero do not affect ozone measurements. The observed temperature behavior closely reflects test conditions in the cold chamber test before the measurement campaign except air temperatures during the flight were lower compared to the test environment.

## 4.4.4 Pressure dependence

The ozone monitor model 205 was successfully operated on airborne platforms in earlier studies by Mynard et al. (2023), Sorooshian et al. (2023) and Yates et al. (2013). In particular the study by Yates et al. (2013) operated the instrument with

similar modifications up to altitudes of 9 km using the Alpha Jet research aircraft as part of the Alpha Jet Atmospheric eXperiment (AJAX) with the 2BTech ozone monitor mounted within an underwing pod. Before their measurement campaign they tested the ozone monitor in a pressure chamber at pressures between 200 and 800 hPa, a similar range as during our TPEx I campaign. Based on these pressure tests no significant pressure dependence could be derived and they reported an overall uncertainty of 3 ppbv for 10 s data. These findings compare quite well with our derived uncertainty for the TPC-TOSS instrument of  $\pm$  (3 ppbv + 0.9 %) for 2 s data. Furthermore, we also did not observe any significant pressure dependency.

## 4.4.5 Collocated performance test

One key objective for the deployed TPC-TOSS was the simultaneous measurement of ozone on the two platforms, the Learjet and TPC-TOSS. We therefore tested the instruments side by side in the laboratory before and on ground during the campaign to identify any systematic error between the two ozone instruments. Based on ambient air measurements in the laboratory before the campaign (Fig. 7a), a difference of  $(0.6 \pm 1.9)$  ppby between the two instruments was observed.

Since the instruments were not in their final setup during the laboratory comparison described before we repeated the side by side comparison during the measurement campaign on ground with both instrument mounted inside the "Knuffi" and inside the TPC-TOSS. This also included the final setup for the inlet tubing providing a calibration of the final flight configuration.

Figure 7. Collocated performance test in the laboratory before the campaign, during the campaign using the final setup as mounted on the respective platforms across different mixing ratios, and in-flight during research flight F10. (a) histogram of  $\Delta O_3$  for the different collocated test environments: laboratory, field and in-flight. The orange shaded area shows the results from laboratory measurements. Coloured lines show the results of the field intercomparison for different mixing ratio steps, the black line shows the overall difference of the field intercomparison. The green shaded area shows the 4 min in-flight intercomparison. For better visibility the values of the individual mixing ratio steps and the in-flight data were multiplied by 5. (b) correlation of the ozone intercomparison at the measurement site.

Figure 8. Colocated performance test in-flight during research flight F10 on 20 June 2024. (a) Time series of ozone measurements and GPS altitude (lower panel) from Learjet (ozone: red, altitude: black) and TPC-TOSS (ozone: blue, altitude: grey) as well as measured outside air temperatures (red colours) and relative humidity over ice (RHI) (green colours) in the upper panel. The dark yellow shaded area shows a time interval when the TPC-TOSS was released at minimum safe rope distance (200 ft  $\approx$  61 m) for around four minutes. At that time the vertical distance between Learjet and TPC-TOSS was around 43 m. (b) histograms of the difference between Learjet and TPC-TOSS data within the yellow shaded time interval.

As shown in Fig. 7a, we performed measurements at four different mixing ratios (20 ppbv, 60 ppbv, 100 ppbv and 140 ppbv) by using the external ozone calibration source. The histograms of  $\Delta O_3$  between both instruments for the individual mixing ratio levels as well as the whole data set of this experiment agree within 2.5 ppbv (1  $\sigma$ ) based on the average difference between both data sets. The maxima of the individual curves deviate by 1 ppbv from  $\Delta O_3$ =0 within the uncertainty range of the instruments derived in Sect. 4.4.2. The correlation between both instruments shows that data points are located along the 1:1 line with a deviation of less than 1 % confirming the laboratory tests. The offset between both instruments of 1.6 ppbv lies within the uncertainty range.

During measurement flight F10 on 20 June 2024 we could perform a quasi co-located test between both instruments (Fig. 8). During this flight, for the reason of in-flight intercomparison of instruments, the retraction of the TPC-TOSS was stopped at around 200 ft (61 m) cable length before the TPC-TOSS was finally attached to the aircraft and switched off. Therefore the TPC-TOSS was measuring for around four minutes at a distance of just 43 m below the aircraft. This offered the possibility for an in-flight intercomparison of the redundant measurements (Fig. 8a). The aerosol size distribution measurement results for this intercomparison are discussed in Sect. 5.3. For ozone the  $\Delta O_3$  histogram (Fig. 7a) for this part of the flight shows a rather broad distribution of around 25 ppby centered around zero since ozone mixing ratios still show some atmospheric variations during the flight (Fig. 8a). The observed offset in the maximum of the distribution of around 5 ppbv is most probably due to the fact that there is still a vertical distance of 43 m between both platforms. For typical vertical ozone gradients of around 600-800 ppbv km<sup>-1</sup> near the tropopause a vertical distance of 50 m would correspond to even 30 ppbv or any smaller value when approaching the tropopause. Middle and right panels in Fig. 8 (b) show histograms of the difference of measured air temperature ( $\Delta T$ ) and relative humidity over ice between Learjet and TPC-TOSS. The observed narrow distribution of ( $\Delta T$ ) peaks around -0.5 K which could probably be explained by a dry adiabatic temperature gradient prevailing in the flight region. A typical gradient of 10 K km<sup>-1</sup> would result in around 0.5 K temperature difference between Learjet and TPC-TOSS at a vertical distance of around 50 m. Assuming a uniform distribution of water vapor mixing ratios in the measurement region, indicated by the uniform distribution of ozone, a higher measured temperature at the TPC-TOSS would result in lower relative humidity values at the TPC-TOSS, which are observed based on the  $\Delta$ RHI distribution (right panel in Fig 8b) that is shifted to slightly positive values.

### 4.5 Aerosol size distribution measurements

For the aerosol measurements, we deployed UHSAS on the TPC-TOSS and the Learjet. These spectrometers are manufactured by Droplet Measurement Technologies (DMT) and measure the size distribution of aerosol particles in the size range between 95 nm and 1000 nm. The measuring principle is based on laser based light scattering in the infrared spectral range. Therefore, UHSAS uses a Nd<sup>3+</sup>:YLiF<sub>4</sub> solid state laser with an operating wavelength of 1054 nm (Cai et al., 2008; Kupc et al., 2018). The laser mode has an intracavity power of approximately 1.1 kW cm<sup>-2</sup> and is perpendicular to the particle stream. Aerosol particles are actively pumped into the detection unit through a jet assembly with a sample flow of 50 cm<sup>3</sup> min<sup>-1</sup> and are focused to a narrow particle beam with a sheath flow. This sheath flow is in the range of 700 cm<sup>3</sup> min<sup>-1</sup> at sea level for the cabin instrument (UHSAS-C) and controlled by a mass flow controller to 600 cm<sup>3</sup> min<sup>-1</sup> for the UHSAS installed on TPC-

TOSS (UHSAS-A). The scattered light is collected by two pairs of Mangin mirrors in the range between 22° and 158° and focused onto the corresponding photodiodes. These photodiodes convert the photocurrent into a voltage which can be assigned to a particle signal by calibration curves.

Figure 9 shows the calculated response of the UHSAS according to Mie theory for a range of refractive indices covering the atmospheric range, similar to Cai et al. (2008) and Mahnke et al. (2021).

Figure 9. Theoretical response of the UHSAS according to Mie-theory and different refractive indices.

## 4.5.1 Size characterization

The assignment of the particle signal into the corresponding size bin is done by calibrating the four gain stages of the photodiodes in the instrument. This calibration is done in two steps, the relative gain calibration and the absolute gain calibration. For the relative gain calibration the instrument needs a broad distribution of particle sizes to determine the coefficients. In contrast to this method, the absolute gain calibration is performed by using polystyrene latex (PSL) particles of known size. In this step, each particle size is assigned to a measured gain value. Typical particle sizes for the absolute gain calibration by the manufacturer are 100, 150, 270 and 500 nm. In order to verify whether the last calibration is valid and both instrument versions operated during the measurement campaign are comparable, we performed a size characterization measurement before and after the campaign. For this, we generated aerosol particles with different refractive indices using an atomizer, dried the aerosol flow with a diffusion dryer and generated a monodisperse aerosol stream by an electrostatic classifier (TSI, Model 3080 including X-ray neutralization of multiple charged particles). For the characterization we use ammonium sulfate, ammonium nitrate, sodium chloride, glucose and PSL. Except for the PSL measurements, we covered the complete size range between 100 and 650 nm in 50 nm steps. For the PSL characterization, we used the sizes 100, 150, 200, 350, 500, 600 and 800 nm. To increase statistics, we performed more than one measurement for most of the sizes up to 450 nm. The results of the characterization measurements are shown in Fig. 10. We observed an offset to smaller diameters in the measurements of both instruments compared to the mobility diameter selected at the classifier and the 1:1 line (Fig. 10a). Depending on the species, the offset varies in the range we expect from the Mie calculations. However, the assignment of the particles into the size bins follows a linear trend with a slope smaller than one and the comparison of both UHSAS systems shows a very good agreement between each other with the results of the size calibrations on the 1:1 line within the noise (Fig. 10b).

We use these calibration measurements, including the shift in diameter and the individual size calibrations for different refractive indices in order to introduce a new bin scheme. Here, we assign the particle signals to less and broader bins to account for the different refractive indices of the particle types. More precisely, we convert the measured 99 bins into 9 bins of quasi-logarithmic spaced channels. This method is also used in an earlier study using this UHSAS-A instrument by Mahnke et al. (2021). For the reassignment to the new bin scheme, we analyze all individual size calibrations and the corresponding particle diameters of the measured size distributions. Furthermore, the new bins are defined to include all data between the 10% and 90% percentile of the measured diameters to account for the uncertainties caused by the different refractive indices.

**Figure 10.** Characterization measurement of the UHSAS systems (-A and -C) for different aerosol particle types; ammonium nitrate (pink), ammonium sulfate (black), sodium chloride (blue), PSL (orange) and glucose (bright blue). The 1:1-line is represented by the grey dashed line. The comparison of the measured UHSAS diameter against the mobility diameter selected by the classifier is shown in (a). The comparison of both UHSAS systems is shown in (b).

## 4.5.2 Counting efficiency

Furthermore, we investigated the counting efficiency of both instruments. For this, we used the same measurements as for the size calibration and compared the total count number, by adding the counts from all size bins. Figure 11 shows the median values for one minute averages of both instruments, for all particle types measured during the laboratory characterization. The data from both instruments show good agreement and are distributed around the 1:1 line. Figure 12 shows the ratio of the median values as a function of size. Here we can see that in a size range between about 300 and 500 nm, the UHSAS-A showed less particle counts than the UHSAS-C, but showed slightly higher particle counts around 100–200 nm. The discrepancies in the range between 300 and 500 nm are unexpected, and there is no clear explanation for them. It may be a combination of imperfect gain calibration and slight laser misalignment causing the undercounting of particles by the UHSAS-A.

**Figure 11.** Median of the counts for a one minute measurement interval for different species, sizes and concentrations. The error bars represent half of the interquartile range between the 75 and 25 % percentile.

**Figure 12.** Ratio between the median counts of UHSAS-A divided by the median counts of the UHSAS-C. The colours and symbols indicate the different aerosol species; ammonium nitrate (pink, circle), ammonium sulfate (black, square), glucose (bright blue, tilted square) sodium chloride (blue, hourglass) and PSL (orange, diamond).

# 5 In-flight performance of TPC-TOSS and atmospheric measurements

### 5.1 TPC-TOSS attitude and position of platforms

During TPEx I in June 2024 the towed sensor shuttle was expected to have a similar or even improved flight characteristic as on previous campaigns (Frey et al., 2009; Finger et al., 2016; Klingebiel et al., 2017). Changes in instrumentation compared to the AIRTOSS-ICE mission in 2013 led to a more symmetric shape of the front of the drag body as the asymmetric CCP-CDP (Cloud Combination Probe - Cloud Droplet Probe) instrument was exchanged with the UHSAS-A instrument with a inlet tube in the center of the circular area of the TPC-TOSS geometry (Fig. 2 lower panel). The ozone bypass inlet and outlet were symmetrically mounted left and right in back part of the TPC-TOSS (Fig. 2 upper panel). In addition, air brakes were installed on the four wings of the drag body to further improve flight behavior.

Figure 13. Excerpt from research flight F10 on 20 June 2024 during TPEx I. The upper panel shows measured mixing ratio of ozone on Learjet (blue) and TPC-TOSS (red) as well as total aerosol number concentration measured with UHSAS-A (yellow). The middle panel contains Learjet (black) and TPC-TOSS (grey) altitude and true air speed of the Learjet (orange). The lower panels show Learjet and TPC-TOSS heading in red and light red and roll and pitch angle of TPC-TOSS in blue and light blue colours. The first, third and fifth dashed rectangle show turns of Learjet and TPC-TOSS and the second and fourth dashed rectangle mark the combination of turn and climb.

Figure 13 shows flight attitude of the TPC-TOSS in different flight phases (turns, climbs and combination of both) during a part of research flight F10. As shown in Fig. 1a the deployment of the TPC-TOSS is only allowed in small restricted air spaces in the North Sea and Baltic Sea with dimensions on the order of 50 x 50 km provoking numerous turns to stay within the air space. Due to the limited operational area, the wire rope length was set to 914 m as mentioned in Sect. 3. This resulted in a horizontal distance between TPC-TOSS and Learjet of 877 ± 3 m on average during undisturbed flight conditions (no turns and no climb or descent). The resulting vertical distance was on average 152 ± 8 m. At ideal stable flight conditions (e.g. F10 7:57:42–8:00:34) the flight behavior is characterized using the following flight parameters. The roll angle of the drag body was stable at -2.43 ± 0.53°, pitch angle average during above mentioned time interval was 0.18 ± 0.16° and heading averaged to -147.45 ± 0.41°. The negative roll angle could be explained by the TPC-TOSS flying slightly sideways from the aircraft which in turn causes an additional force component on the TPC-TOSS to the left from the towing cable (Frey et al., 2009).

The lateral distance between TPC-TOSS and Learjet was on average 89 ± 8 m based on flight F10. Pitch angle and heading 465 stayed very stable during undisturbed flight conditions. Deviations from this flight conditions could be introduced by turns (changes in the heading accompanied by changes in the roll angle) and climbs (additional change in the pitch angle) as the TPC-TOSS is slightly accelerated during climb resulting in the nose moving down which in turn gives small positive deviation of the pitch angle. Furthermore, stronger variations in the roll angle at constant flight conditions with respect to turns and 470 climbs or descents could be forced by turbulence in the atmosphere. As flights were designed to study the effect of turbulence introduced by for example internal dynamics of cirrus clouds on the chemical composition of the atmosphere it was expected to experience these types of flight conditions. Fortunately, none of the deviations from stable flight conditions affected trace gas and aerosol measurements significantly as shown in the upper panel of Fig. 13. In particular during turns and altitude changes measurements seem unaffected from a significant change in the roll angle during flight manoeuvres. In contrast to previous campaigns that needed a stable roll angle of the TPC-TOSS within  $\pm 3^{\circ}$  for proper radiation measurements, trace species and 475 aerosol measurements during TPEx I were still possible in turns and climbs and the decay time (time needed for the TPC-TOSS to recover to stable attitude conditions after turns and/or climbs and descents) could still be used as measurement time. In addition temperature and humidity measurements from the ICH sensor on the TPC-TOSS (not shown) seem unaffected during the decay times.

## 480 5.2 Meteorological parameters and ozone measurements

The measurements during TPEx I provide simultaneous and co-located in situ measurements of aerosol and ozone at a short vertical distance between the TPC-TOSS and the Learjet. This enables, for the first time, the determination of gradients of these substances in the UTLS based on in situ data. The simultaneous measurements are in particular important for studying the effect of transient small scale dynamics in the UTLS on the composition of and mixing in the respective region. Features associated with small scale turbulent dynamics are often very short in time and limited in space. This makes them often difficult to probe sufficiently with only measurements from an aircraft. Determination of turbulent fluxes require the observations of gradients (e.g. Shapiro, 1980). With the TPC-TOSS this is now possible without correcting for larger time lags due to multiple necessary legs of the aircraft through a turbulent region. However, also with the TPC-TOSS there is lag which needs to be considered, since the TPC-TOSS is up to 900 m behind and up to 100 m sideways of the aircraft. Still, this lag is much smaller than the lag associated with aircraft only measurements and amounts to 5–6 s. The temperature measurements combined with the pressure data provide potential temperature gradients in the UTLS region. This in turn provides a measure of stability in the respective altitude range covered by TPC-TOSS and Learjet. Together with the vertical gradients of ozone and aerosol we can study the effect of changes in stability (triggered for example by clouds) on the composition of the UTLS and also mixing processes.

The advantage of the dual platform approach during TPEx I is summarized in Fig. 15. Figure 14a shows the corresponding flight track of research flight F03 on 11 June 2024 as well as the altitude profile of the flight. The target region was the restricted air space in the Baltic Sea. The aim of that flight was to probe mixing in the upper troposphere and tropopause region within an area of low tropopause altitudes. The flight was planned with stacked flight levels of 1000 ft (≈ 305 m) distance within

**Figure 14.** (a) Flight track of research flight F03 during TPEx I on 11 June 2024. The flight track is colour coded with altitude as also given by the inset figure. (b) Interpolated potential vorticity from the ECMWF IFS model along the flight track between 500 hPa and 275 hPa. The map was created from public-domain GIS data found on the Natural Earth website (http://www.naturalearthdata.com, last access: 30 June 2025).

the restricted air space after the TPC-TOSS was deployed. Figure 14b shows the potential vorticity from the ECMWF IFS (European Centre for Medium Range Weather Forecast - Integrated Forecast System) along the flight track between 275 hPa and 500 hPa. While the early and late parts of the flight are in the troposphere, the Learjet ascended stepwise deeper into the stratosphere during the stacked flight levels.

Figure 15 shows time series of different quantities of that part of the flight when TPC-TOSS was released. The top panel shows the heading of the TPC-TOSS in grey indicating the frequent turns that were flown in the restricted air space. The altitude difference between Learjet and TPC-TOSS is shown in black. On average the TPC-TOSS was located 170 m (range 130 to 200 m depending on flight condition) below the aircraft. Short periods of time with an increasing vertical altitude distance up to 200 m between Learjet and TPC-TOSS were only observed during the short climbs to the next flight level. The middle panel shows the difference of ozone (blue) and potential temperature (red) between Learjet and TPC-TOSS derived from co-located measurements of ozone and temperature on TPC-TOSS and Learjet. The lower panel shows the determined gradients of ozone (green) and potential temperature (orange) with altitude. The shaded areas in this figure denotes the errors of the measured and calculated quantities. The altitude is measured with an uncertainty of  $\pm$  2 m, the uncertainty for the temperature measurements is  $\pm$  0.32 K. For the calculation of potential temperature the uncertainty of the determined pressure value of  $\pm$  1 hPa needs to be further considered. Also taking into account the uncertainty for ozone ( $\pm$  (3 ppbv + 0.9 %) for TPC-TOSS and  $\pm$  (3 ppbv + 0.7 %) for "Knuffi") these individual errors propagate into the determination of the gradients resulting in an uncertainty of the ozone gradient of up to 10 % and for the potential temperature gradient ( $\Theta$ ) the uncertainty amounts to 31 %.

The  $\Theta$  gradient could now be used to get an indication of stability in the atmosphere. During the first part of the flight with TPC-TOSS released until around 11:25 UTC almost no vertical gradient in ozone and  $\Theta$  is observed indicating a flight mostly in tropospheric conditions (Fig. 14b). The stratosphere in general is characterized by strong static stability and thus a positive

**Figure 15.** Time series of the altitude difference between Learjet and TPC-TOSS including Learjet heading (upper panel), difference of ozone and potential temperature between both platforms (middle panel) and ozone and theta gradients along the flight (lower panel). The shading in all three panels denotes the total uncertainty of the shown parameter consisting of individual measurement uncertainties of the respective parameters. For ozone related quantities the uncertainty is much smaller than the observed variability and thus hardly seen on the figure. The yellow boxes indicate the time interval of the climbs to the next flight level.

potential temperature gradient. Ozone is strongly increasing with altitude in the stratosphere also resulting in a positive vertical gradient. The increasing vertical gradient in  $\Theta$  and ozone from 11:25 UTC on until 11:45 UTC thus indicates increasing stratospheric influence. Afterwards there are strong variations observed in the gradients. Reduced vertical gradients in potential temperature could indicate a less stable stratification of the atmosphere with a potential for mixing which in turn could further reduce the vertical ozone gradient as for example observed between around 11:45 and 11:50 UTC. While a further analysis of  $\Theta$  and ozone gradients is beyond the scope of this paper, the example of research flight F03 shows the potential of the dual platform approach to identify and study mixing processes in the UTLS regions.

### 5.3 Aerosol size distribution measurements

For the in-flight performance analysis and comparison of both UHSAS systems at the TPC-TOSS and Learjet we use an interval during flight F10 (20 June 2024). Here, both platforms were operated on the shortest possible vertical distance of about 43 m for several minutes. We averaged the number, surface, and volume size distribution from both UHSAS systems over 60 seconds (Fig. 16). Also shown in Fig. 16 are the size distributions measured by the Sky-OPC operated in the Learjet cabin. For this time period, we observe a very good agreement of both UHSAS systems, especially for particles smaller than 500 nm. For particles larger than 500 nm, there are some differences between UHSAS-A and -C, especially in the surface and volume distributions. This difference is likely explainable by different gain stitching of the instruments. The comparison for particles larger than 250 nm with the SkyOPC basically shows a good agreement, but the SkyOPC overestimates particle surface and volume between 250 and 300 nm compared to both UHSAS.

**Figure 16.** Timeseries of the Learjet and TPC-TOSS altitude in combination with the STP corrected number concentrations of both UHSAS (-A and -C) during the last minutes in the restricted airspace of F10 on 20 June 2024 between 09:26:00 and 09:27:00 UTC. (a). The orange box marks the period with the TPC-TOSS at the closest safe operation distance to the Learjet (ca. 43 m) for instrument in-flight intercomparison. The averaged number (b), surface (c) and volume (d) distributions of both UHSAS and the SkyOPC in the Learjet cabin are shown in the subpanels.

#### 6 Summary and conclusion

During TPEx I, we demonstrated the first closely co-located, simultaneous trace gas and aerosol measurements in the UTLS region using a tandem platform approach, consisting of a Learjet 35A and a redesigned towed sensor shuttle, TPC-TOSS.

TPC-TOSS was positioned between 95 and 220 m below, up to 900 m behind and up to 100 m lateral to the aircraft during flights.

Based on laboratory and field intercomparisons, redundant instrumentation for temperature and ozone measurements on TPC-TOSS and Learjet agree better than 0.45 °C for the temperature data (based on the individual total uncertainties of the respective ICH sensors of 0.32 °C). For ozone the agreement is better than 4.2 ppbv + 1.1 % for the range of ambient ozone measurements (based on total uncertainties of the individual ozone instruments). In addition also aerosol size distribution measurements by the UHSAS instruments on both platforms indicate similar structures for specific flight sections, which is however partly masked by the high natural variability of aerosol number concentration.

Our results showed that the dual platform approach with the instrument performances reported above in particular allows for using measured temperature gradients as an indication for static stability and to further derive ozone gradients based on the simultaneous measurements at two altitudes in the UTLS. Comparing both, ozone and  $\Theta$  simultaneously at two levels allows to identify the effect of diabatic ( $\Theta$  changing) processes on the ozone distribution and thus mixing. As an example during research flight F03 (Fig. 15) variations in static stability based on  $\Theta$  gradients could either be linked to more tropospheric influence or diabatic processes.

In addition to trace species measurements, two UHSAS instruments deployed on TPC-TOSS and the Learjet provide, for the first time, the opportunity to study the impact of small-scale dynamical features on aerosol concentration and size distribution in the UTLS. A recent study by Joppe et al. (2025) further exploited the potential temperature gradient derived from the dual-platform measurements to analyze the radiative impact of biomass-burning aerosol transported into the UTLS by warm conveyor belt transport.

While a more detailed analysis of trace species and aerosol behavior is beyond the scope of this paper, this study highlights the advantages of the successful deployment of the novel re-designed TPC-TOSS providing a unique data set well suited for the analysis of transient small scale dynamics on the UTLS composition during TPEx I in 2024 and future campaigns.

Data availability. In situ data are available from the Zenodo platform (https://doi.org/10.5281/zenodo.15371527, (Lachnitt, 2025)). Data from ECMWF for the IFS forecast has been retrieved from the MARS system (https://confluence.ecmwf.int/display/UDOC/MARS+user+documentation). Data description is available from https://www.ecmwf.int/en/forecasts/datasets/set-i.

Author contributions. Author contributions. HB wrote the paper, with significant conceptual input from PH, PJ and YL and critical feedback from all co-authors. HB, JS, PJ, NE, AB, SI, SR, CR, AA and LS operated instruments in the field and analyzed resulting data. DK provided IFS data along the flight path. HCL provided the merged data file for analysis. NE and PH are responsible for ozone, GNSS/INS and UMAQS data. AA and CR are responsible for NIXE-CAPS, FISH and WASUL data. PJ and JS are responsible for UHSAS, SkyOPC and CARIBIC-AMS data. YL and AP are responsible for ICH and BCP data. SR und JC are responsible for MC-CPC data. AB and AV are responsible for the SOAP data. SI, KK and LS are responsible for the impactor samples and the analysis in the SEM. AK, TR and SH were responsible for planning, manufacturing and certifying the TPC-TOSS.

Competing interests. Some authors are members of the editorial board of the journal Atmospheric Measurement Techniques.

Acknowledgements. All authors acknowledge the team of enviscope GmbH and GFD GmbH for the opportunity to carry out the campaign and the technical support during the campaign.

All Authors acknowledge funding by the Deutsche Forschungsgemeinschaft (DFG, German Research Foundation) – TRR 301 – Project-ID 428312742: "The tropopause region in a changing atmosphere"

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
