# Peer review of "The TropoPause Composition TOwed Sensor Shuttle (TPC-TOSS): A new airborne dual platform approach for atmospheric composition measurements at the tropopause"

_EGUsphere, 2025_

## Author Comment (AC1)

EGUSPHERE-2025-3175

**Review of the manuscript „The TropoPause Composition TOwed Sensor Shuttle (TPC-TOSS): A new airborne dual platform approach for atmospheric composition measurements at the tropopause"**

by Bozem et al.

Reply to referee #1

We appreciate the kind words on our manuscript and thank the referee for the constructive comments and proposed suggestions. These helped to improve the manuscript. We will answer to all comments of referee #01 below point by point. Referee comments are given in standard, answers in red, and changes to the manuscript in blue font.

This is a nice manuscript describing the use of a novel sensor package that is towed behind an aircraft on a cable. Case studies that demonstrate the use of this technique in the tropopause region where there can be very significant but transient structures that produce strong vertical gradients are presented. The manuscript is well written and generally clear, and the subject matter is appropriate for publication in AMT. Minor revisions are needed to address a few questions, correct technical issues, and provide additional information.

1) Table 2: Please provide a column for instrument uncertainties at the stated sampling frequency. I believe the SkyPOC particle size range is misstated; Bundke et al report a lower detection limit of 0.25 µm.

Thank you for the suggestion. In Table 2-4 an additional column was added stating the uncertainty of the respective instrument where applicable. Thank you for pointing out the wrong size range. This was a typo, the correct lower detection limit is 250 nm. The text in the table was changed accordingly.

2) Section 3. At some point here I would like a brief discussion of how the cable system works and how far the TPC-TOSS module can be lowered. What is the total cable length and the typical vertical separation? This is evident only in graphs. What is the range of deltaZ (or cable length) that could be used safely? Alternatively this could go in Section 5.1.

Following the suggestion of both reviewers we added the following paragraph about the handling of the winch and TPC-TOSS as well as wire length in section 3:

The TPC-TOSS is attached to a winch under the right aircraft wing that is equipped with a steel wire of a maximum length up to 4 km. The pilots operate the winch to release the drag body to the desired wire length and retract it after the measurements. For certification reasons the operation of the winch is only allowed below 25000 ft (7.6 km)

while the maximum flight altitude with the TPC-TOSS deployed is 41000 ft (12.5 km). During the TPEX I flights with the TPC-TOSS a wire length of 3000 ft (914 m) was used. The main reason for not using a longer wire length was the military controlled restricted air space with a maximum side length of 50-80 km in which we were only allowed to fly with TPC-TOSS due to safety constraints. The small area resulted in multiple turns during aircraft operation. Based on the experience from earlier campaigns in the same airspace, the chosen wire length was a compromise between a maximum reachable vertical distance between Learjet and TPC-TOSS and safe and feasible Learjet operation (Klingebiel et al. (2017, and references therein). With this wire length a vertical distance between Learjet and TPC-TOSS of 152 ± 8 m was reached during stable flight conditions (no turns or climbs/descents). The maximum range of vertical distance was between 95 m and 220 m including turns and altitude changes. Further details on the relative position of TPC-TOSS and Learjet are discussed in Sect. 5.1.

In Sect. 5.1 we added the following for more details on the relative position between Learjet and TPC-TOSS.

Line 437 ff.: Due to the limited operational area, the wire rope length was set to 914 m as mentioned in Sect. 3. This resulted a horizontal distance between TPC-TOSS and Learjet of 877 +- 3 m on average during undisturbed flight conditions (no turns and no climb or descent). The resulting vertical distance was on average 152 +- 8 m.

Line 444 ff.: The lateral distance between TPC-TOSS and Learjet was on average 89 +- 8 m based on flight F10.

3) Line 214. Change to, "the addition of insulation to protect the instrument by maintaining temperatures above 0 degrees C." Is this an arbitrary temperature limit or would the ozone instrument still function at colder temperatures? I ask because tropical missions near the tropopause might see much colder temperatures than this (if ~13-14 km altitude could be reached).

We changed the text accordingly. With the "cold weather upgrade" provided by 2BTech, which we applied, the lower limit of the operating temperature range for the ozone instrument is -20°C. We aimed at keeping the temperature inside the instrument above freezing point to avoid any ice formation. Notably, the TPC-TOSS instrument had to be switched on under cold environmental conditions since it was not allowed to have the TPC-TOSS powered while attached to the Learjet. Insulation helped to reduce the cooling rate of the ozone monitor before switching it on 15-30 min after takeoff.

4) Figure 6b. What is the standard deviation of the Gaussian fit? This would inform as to total instrument variance.

The standard deviation of the Gaussian fit is 1.6 ppbv and is thus on the order of the noise determined with regular calibrations in the field. We added the following sentence to the discussion of Fig 6b:

The standard deviation of the gaussian fit amounts to 1.6 ppbv and is on the order of the instrument noise determined in the previous section.

5) Line 343. Surprising use of imperial length units. I thought this was strictly a problem in the U.S.!

Thank you for the hint. We added the length in SI units.

6) Line 354 "peeks" -> "peaks" and line 302 "week" -> "weak".

Correct. We changed it accordingly

7) Line 382 "Atomizer" -> "atomizer"

Correct. We changed it accordingly

8) Line 395. Reference to Fig. 17 before Fig. 12. Generally figures need to be cited in order. You could place the bin diameters as vertical lines in Fig. 11b instead, if you prefer. This might help see how they span the range of compositions. Except, see the next comment below.

We removed the reference to Fig. 17 at this place, as this reference might anyway be misleading, because Fig. 17 does not show the exact new bin scheme but only a size distribution within the new bin scheme.

9) Fig. 11. There is a substantial (~20%) shift in diameter from the manufacturer's calibration, consistently for both instruments. It's not clear if this correction has been applied when the new wider bins were created. I'm not sure of the reason for creating the wider bins, other than some way to represent the range of possible sizes. It may be better to calculate a low-refractive index calibration and a high-refractive index calibration (by calibration, I mean relationship between channel number and calibration diameter for each calibrant), then a medium-refractive index calibration as the default value. Uncertainty bars would then span across the low- and high- refractive index cases and you could still use the full 99-channel resolution of the UHSAS. I'm not sure what the wider bins gains you since using that method a central bin diameter is assumed and only one size distribution, with no uncertainty range, comes out. Uncertainty ranges might be more useful than grouped wider bins.

The shift in diameter from the manufacturer's calibration to our calibration measurements has been taken into account before adjusting the bin schemes.

We performed a low-refractive index and a high-refractive index calibration: The low-refractive index calibration was done with glucose and the high-refractive index calibration with PSL. We chose to reduce to fewer and wider bins according to an earlier study using exactly this UHSAS-A instrument (Mahnke et al., 2021, ACP). Here, we did not just group the bins together, but we also looked at the statistics to account for the data within the 10% and 90% percentiles. Further, we think that the readability of size

distributions is easier with a reduced number of size channels compared to a large number of size bins with individual uncertainties.

We added the following text to section 4.5.1:

We use these calibration measurements, including the shift in diameter and the individual size calibrations for different refractive indices in order to introduce a new bin scheme. Here, we assign the particle signals to less and broader bins to account for the different refractive indices of the particle types. More precisely, we convert the measured 99 bins into 9 bins of quasi-logarithmic spaced channels. This method is also used in an earlier study using this UHSAS-A instrument by Mahnke et al. (2021). For the reassignment to the new bin scheme, we analyze all individual size calibrations and the corresponding particle diameters of the measured size distributions. Furthermore, the new bins are defined to include all data between the 10 % and 90 % percentile of the measured diameters to account for the uncertainties caused by the different refractive indices.

10) Fig. 13. There are some surprising size-dependent counting efficiency differences between units here, which pass without much comment. ~30% is a big counting difference (i.e., 350 nm). What is going on? Any ideas?

We agree that these differences are unexpected. We are not sure what has caused them, but they could be due to imperfect gain stage calibration combined with laser misalignment, resulting in reduced sensitivity in this size region. The laser had to be replaced before the campaign and, as the official support for the UHSAS-A has been discontinued by the manufacturer, the new alignment was carried out to the best of our ability. We added the following sentence in the discussion of Fig. 13:

The discrepancies in the range between 300 and 500 nm are unexpected, and there is no clear explanation for them. It may be a combination of imperfect gain calibration and slight laser misalignment causing the undercounting of particles at the UHSAS-A

11) With PSL, when you are comparing numbers do you just integrate the PSL peak, or are you counting additional surfactant/contamination particles in the smallest bins (assuming no DMA is used for the PSL calibrations to remove the smaller contaminant particles)?

For all measurements including PSL, we used the DMA to remove contaminant particles. For all number concentrations we integrated over the region around the selected size peak for all calibration substances.

12) Line 417. The yaw angle (alignment with respect to the local wind vector) of -147 degrees must be an error. I might believe -1.47 degrees.

We used the wrong term since we refer to "heading" here, as also shown in Fig. 14. The text was corrected accordingly.

13) Fig. 16. What is the shading on this plot?

The shading in figure 16 denotes the total uncertainty of the shown parameters calculated from individual uncertainties of measured parameters used to derive for example ozone and Theta gradients (lower panel in Fig. 16). The caption was changed accordingly:

The shading in all three panels denotes the total uncertainty of the shown parameter consisting of individual measurement uncertainties of the respective parameters. For ozone related quantities the uncertainty is much smaller than the observed variability and thus hardly seen on the figure.

14) Line 483. Two periods after "cabin".

Correct. We changed it accordingly.

15) Figure 17. I don't find log-log size distributions very useful. Of more interest (at least to me) would be how well the integrated number, surface, volume, and effective radius agree. These are the parameters governing CCN activity, heterogeneous chemistry, extinction and mass transport, and remote sensing retrieval, respectively.

See reply to 16.

16) If data need to be plotted on a log axis, it implies that the parameter is not normally (Gaussianly) distributed. Thus standard deviation, which assumes Gaussian statistics, is not valid and is meaningless in describing the statistics. A geometric standard deviation might be better here. (But I would prefer linear plots of N, S, and V vs log diameter instead.)

We chose the log axis in order to see also smaller numbers in the particle diameters. However, we agree with your comment and according to your suggestions we added N, S, and V as linear plots vs. log diameter.

17) The lateral and fore-aft spacing of the TPC-TOSS is mentioned in Section 6, but of more importance is the vertical spacing, which is not mentioned.

As stated for point (2) we added more information on the vertical spacing in Sect. 3. We further added the information on vertical spacing in Sect. 6 and modified the sentence as follows:

TPC-TOSS was positioned between 95 and 220 m below, up to 900 m behind and up to 100 m lateral to the aircraft during flights.

18) Please make sure that all figures are plotted using colors and/or symbols that would allow a person with a color vision impairment to distinguish the different parameters. There are two such scientists in my close acquaintance and it can be a struggle for them.

We checked and modified figures to our best ability to account for a color vision impairment. In case further changes are necessary we will modify the figures accordingly.

References:

Mahnke, C., Weigel, R., Cairo, F., Vernier, J.-P., Afchine, A., Krämer, M., Mitev, V., Matthey, R., Viciani, S., D'Amato, F., Ploeger, F., Deshler, T., and Borrmann, S.: The Asian tropopause aerosol layer within the 2017 monsoon anticyclone: microphysical properties derived from aircraft-borne in situ measurements, Atmospheric Chemistry and Physics, 21, 15259–15282, https://doi.org/10.5194/acp-21-15259-2021, 2021.

---

## Author Comment (AC2)

EGUSPHERE-2025-3175

**Review of the manuscript „The TropoPause Composition TOwed Sensor Shuttle (TPC-TOSS): A new airborne dual platform approach for atmospheric composition measurements at the tropopause"**

by Bozem et al.

Reply to referee #2

We appreciate the kind words on our manuscript and thank the referee for the constructive comments and proposed suggestions which helped to improve the manuscript. We will answer to all comments of referee #02 below point by point. Referee comments are given in standard, answers in red, and changes to the manuscript in blue font.

The article presents a very interesting new platform for airborne measurements at the tropopause altitude range. It is well-written, and fully fits in the range of the journal AMT. In my opinion it can be published after some minor revisions. Suggestions for improvement and small typos are specified below.

Suggestions for improvement:

- I would suggest to introduce the method earlier, e.g. the first figure should be a sketch of how it works, with the Lear Jet, the rope and the payload. In my opinion it takes too long for the reader to get a first impression in Fig. 3/ on page 8. This should also include a clear statement if there is only a mechanical connection, or also power supply

  Following this suggestion, we added Fig. 1b showing a schematic of the concept of the dual platform approach and extended the caption of Fig. 1 as follows:

  (b) Schematic of the concept of the dual platform approach with TPC-TOSS attached to the Learjet aircraft with a steel wire rope allowing for simultaneous measurements at two levels. Colors in the background represent an arbitrary air mass property changing from low to high values at the tropopause. This property can be measured simultaneously by the two platforms. Modified from Emig et al. (2025).

  Furthermore, we added the following sentence in Sect. 1: The TPC-TOSS was attached to the aircraft via a purely mechanical connection using a steel wire rope.

- Please state on the swinging behaviour of the system, e.g. show statistics on pitch/roll/yaw angles during one flight, mention critical situations, describe more

in detail how the tethered system is handled, e.g. with a winch. There is some information in the summary (900 m behind, 200 m lateral) – how constant is this?

The TPC-TOSS is flying quite stable during undisturbed flight conditions (without turns and altitude changes). During these conditions the following statistics are given as mean values and standard deviation over flight F10 (see figure below):

- Altitude difference between Learjet and TPC-TOSS: 152.3 +- 8.3 m
- Pitch angle: -0.02 +- 0.60 °
- Roll angle: -2.52 +- 2.35 °
- Horizontal distance between Learjet and TPC-TOSS: 877.2 +- 3.1 m
- Lateral distance between Learjet and TPC-TOSS: 88.7 +- 8.2 m

[Figure]

Figure 1: Time series of different attitude and position parameters during research flight F10 on 20 June 2024 during TPEX I. The upper panel shows relative positions of the TPC-TOSS to the Learjet. The lower panel shows heading (red), roll (blue) and pitch (green) angles of TPC-TOSS and Learjet (only heading) during the flight.

Deviations from these undisturbed flight conditions occur during altitude changes and turns. Altitude changes of Learjet and TPC-TOSS are indicated by a higher pitch angle during climb or a lower pitch angle during descent in Fig 1., turns are indicated by a change in the Learjet and TPC-TOSS heading. These are accompanied by significantly higher roll angles of the TPC-TOSS following the

turn until it is in stable flight mode after a few seconds. For the particular flight larger variations in the roll angle along part of the horizontal flight sections are visible. These are mainly caused by turbulence occurrence during the flight. Notably, none of these deviations from undisturbed flight conditions significantly affect trace gas and aerosol measurements as discussed in Fig 14 in the preprint.

Critical situations during the flight in general arise from turns as these add additional force to the rope and too sharp turns might lead to a rupture in the wire rope.

To further analyse the oscillating behaviour of the TPC-TOSS we performed frequency analysis of the attitude parameters of TPC-TOSS but could not identify regular oscillations in these parameters. There are some flight intervals for which the frequency analyses indicate oscillations with a period of around 20 s of unspecific origins. However, measurements were not affected by these oscillations.

[Figure]

Figure 2: Same as Fig. 1 with a zoom on a specific time interval.

With respect to the handling of the towed sensor shuttle we added the following paragraph in Sect. 3:

The TPC-TOSS is attached to a winch under the right aircraft wing that is equipped with a steel wire of a maximum length up to 4 km. The pilots operate

the winch to release the drag body to the desired wire length and retract it after the measurements. For certification reasons the operation of the winch is only allowed below 25000 ft (7.6 km) while the maximum flight altitude with the TPC-TOSS deployed is 41000 ft (12.5 km). During the TPEX I flights with the TPC-TOSS a wire length of 3000 ft (914 m) was used. The main reason for not using a longer wire length was the military controlled restricted air space with a maximum side length of 50-80 km in which we were only allowed to fly with TPC-TOSS due to safety constraints. The small area resulted in multiple turns during aircraft operation. Based on the experience from earlier campaigns in the same airspace, the chosen wire length was a compromise between a maximum reachable vertical distance between Learjet and TPC-TOSS and safe and feasible Learjet operation (Klingebiel et al. (2017, and references therein). With this wire length a vertical distance between Learjet and TPC-TOSS of 152 ± 8 m was reached during stable flight conditions (no turns or climbs/descents). The maximum range of vertical distance was between 95 m and 220 m including turns and altitude changes. Further details on the relative position of TPC-TOSS and Learjet are discussed in Sect. 5.1.

And we added the following sentences to Sect. 5.1:

Line 437 ff.: Due to the limited operational area, the wire rope length was set to 914 m as mentioned in Sect. 3. This resulted in a horizontal distance between TPC-TOSS and Learjet of 877 +- 3 m on average during undisturbed flight conditions (no turns and no climb or descent). The resulting vertical distance was on average 152 +- 8 m.

Line 444 ff.: The lateral distance between TPC-TOSS and Learjet was on average 89 +- 8 m based on flight F10.

- There are different informations about altitude, e.g. in the intro it says 6-12 km. This is a contradiction to l. 69, studying vertical transport form the PBL into the UTLS. Then in l 84 it states that the maximum altitude with the TPC-TOSS was only 9700 m.

  We corrected inconsistent altitude information given in the paper. In general, during the TPEx I mission 8 scientific measurement flights were performed. During 4 out of these 8 flights the TPC-TOSS was deployed. It is important to note that the Learjet allows for a high flexibility for operating the TPC-TOSS or not. The drag body including the winch system can be removed between flights. Depending on scientific questions for a specific research flight the winch system including TPC-TOSS was attached to the aircraft or not. Therefore, half of the scientific flights were flown without the TPC TOSS and the others with the dual platform. For the flights without the TPC-TOSS, we probed the atmosphere from

ground levels up to an altitude of 12 km and with the TPC-TOSS deployed we covered altitudes between 6.4 and 10.9 km.

- Different informations about aerosol sizes: 95 nm-1 µm in l. 77

The information given in line 77 refers to the size range of the UHSAS instruments operated on both platforms, Learjet and TPC-TOSS. The optical particle counter measuring up to 3 µm was only operated on the Learjet.

- Please motivate more in detail the 200 m rope length. Was this a choice based on technical constraints or scientific scales? In both cases please explain more in depth. L. 343 states that the rope was only 200 ft – is this flexible? Can it be chosen for each flight?

To derive gradients of trace species, aerosol and meteorological parameters, we equipped the Learjet and the TPC-TOSS partly with similar instruments for which we did intercomparison measurements in the lab and in between research flights on the ground as discussed in Sect. 4. To have an in-flight comparison of the redundant instrumentation on TPC-TOSS and Learjet we aimed for a part of the flight with a minimum distance between TPC-TOSS and Learjet. For safety reasons, the TPC-TOSS is switched off while attached to the aircraft and the winch system. During release of the TPC-TOSS it takes a few minutes until all instruments are working properly and thus the distance between Learjet and TPC-TOSS is already large. On research flight F10 we took the opportunity to stop retracting the TPC-TOSS at a minimum safe distance to fly for four minutes in that configuration. This minimum safe distance corresponded to a wire length of 200 ft (61 m) which in turn resulted in a vertical distance of 43 m and allowed for a quasi colocated performance test of the instrumentation as described in Sect. 4.4.5.

In general, the rope length is flexible between the minimum length allowed to safely operate the aircraft and the maximum length of 4 km. As discussed in the additional and new paragraph in section 3 the rope length of 914 m during the research flights was a compromise between a maximum reachable vertical distance between Learjet and TPC-TOSS and feasible operation of the dual platform configuration in the small, restricted air spaces. In addition, operating the winch is only allowed below 25000 ft so that the length of the rope is once set at the beginning of the research flight and cannot be changed during TPC-TOSS operation at higher altitudes. For all flights with TPC-TOSS deployed we kept the rope length at 914 m.

- If relevant, please explain quickly the Mission Support System, or omit.

The Mission Support System (MSS) is an important tool for detailed flight planning, in particular for flights with TPC-TOSS deployed. We extended the section on MSS as follows:

For operational planning of the flights we used the Mission Support System (MSS, Bauer et al. (2022)) with meteorological and chemical data from ECMWF from the IFS and CAMS forecast models. MSS as a server client application allows to interactively plan flight trajectories based on current four dimensional forecast data. Additionally, we used high resolution data from the ICON-D2 for forecasts of convection as well as from ICON for WCB forecasts.

- Please include technical details on temperature management. The aerosol sensors are for sure temperature stabilized? How cold does it get in the TPC-TOSS without heating, how much heating power is applied? Is it actively controlled depending on measured inside temperatures? L. 231 only mentions that the system is thermally isolated

For the TPC-TOSS there is no active temperature management of the drag body itself or any instrument inside due to limitations in available power from the battery pack. Instruments and the drag body are only heated from the heat the instruments produce during operation which in turn results in a higher temperature within the drag body in comparison to environmental temperatures outside. We did not measure the temperature of the drag body volume itself but had temperature measurements inside the ozone instrumentation for example. Based on former measurement campaigns minimum temperatures inside the drag body reached -20 to -25 °C at the time instruments were switched on. The ozone instrument was certified to operate only above -20°C, all other instruments and components are certified for lower temperatures. Therefore we installed BASOTECT foam around the ozone instrument for a passive insulation to reduce the cooling rate of the instrument during the flight time before switching on the TPC-TOSS when all instrumentation inside the TPC-TOSS had to be switched off. Details on the temperature evolution of the ozone instrument are discussed in Sect. 4.4.3.

- 150/151: the uncertainty is 1.25 and 2 m. Is this good enough? Please comment.

The uncertainty of 1.25 m (horizontal position) and 2 m (vertical position) for the determination of the position based on the GNSS/INS instrumentation resulted in relative errors of 0.1 % (horizontal) and 1.3 % (vertical) taking into account the average horizontal and vertical relative distance between Learjet and TPC-TOSS. The error of the position is included in the uncertainty of the gradients discussed in Fig. 16 (shaded region for the respective parameter) by applying Gaussian error propagation. Since relative errors of the position are in the same order of magnitude or even smaller than instrument uncertainties for the respective

parameters this uncertainty allows to determine ozone or temperature gradients of a few 100 ppbv / km.

- 167/168: what is the temporal resolution of the humicap in the UTLS? A few minutes would be too much for the scientific questions, I suppose? Why not complement with an optical hygrometer?

  Thank you for bringing up this point. The data output frequency of the humicap as part of the ICH sensor regularly operated within the IAGOS framework since 2011 is 1 Hz. However, the response time of the humidity measurement of 1 s (altitude region 0-3 km) is reduced to 15 s (altitude region 3-6 km) and up to 180 s (altitude region 6-12 km) based on Rolf et. al. 2024 (and references therein). During TPEX I, in the Learjet cabin, the FISH instrument was operated serving as a reference instrument for humidity measurements in particular in the UTLS as shown during previous campaigns (DENCHAR, AIRTOSS I + II) with the Learjet (Rolf et al., 2024). Based on humidity intercomparisons during these campaigns the agreement between FISH and the ICH sensor is 9 % for water vapor concentrations in the range 30-300 ppmv and better in the range 300-1000 ppmv. It will of course depend on the scientific question in which way the humidity data from the ICH sensor can be used. While small scale fluctuations of humidity in the UTLS might not be resolved with the ICH data, a valuable insight in the water vapor distribution in the UTLS using the two-platform approach is still possible but outside the scope of this paper.

  For the TPC-TOSS measurements an additional humidity instrument would of course be beneficial but almost impossible to realize due to limitations in space and weight.

- In general, how do you address the issue of response time? What corrections are applied? Maybe compare to the correction methods applied in Bärfuss et al., 2023 (https://amt.copernicus.org/articles/16/3739/2023/amt-16-3739-2023.html), who performed temperature and humidity measurements up to 10 km altitude based on a drone

  There are no corrections applied to the humidity data of the ICH sensor with respect to the response time. The aforementioned increased response time of ICH humidity with decreased temperature/altitude was derived for the data quality assurance of ICH operation during MOZAIC and IAGOS by Neis et al. (2015), who applied an exponential moving average (EMA) to the reference data from the Fast In-situ Stratospheric Hygrometer (FISH), which was measuring in parallel to the ICH sensor on earlier Learjet field campaigns. As noted in the reply to the last comment, the ICH sensors are a compromise made to observe the horizontal distribution of humidity at stable flight levels and possibly the vertical difference in water vapour concentration/humidity between two platforms. The

small-scale fluctuations of humidity, especially during the ascent and descent legs, if necessary, might be seen in ICH by trying numerically deconvolution of the measured signal based on the response time, which is beyond the scope of this paper.

- Explain colours of figures only in captions, do not use in text, e.g. l. 297-299, 330, 346, 413

  Thank you for the hint. We changed it accordingly.

- 7: were temperature corrections applied, similar to Bärfuss et al.? If yes, please explain method in text. If not, why? What error does this imply?

  The time lag of Pt1000 clarified in Bärfuss et al. (2023) is an interesting point. After careful consideration, we don't think it has a big impact on the temperature readings of the Pt100 (1 Hz) and therefore, a similar spectral correction was not applied here given the scope of the TPEx I campaign and the application cases of the ICH sensors. However, the adiabatic heating in the Rosemount inlet housing, which causes a substantial temperature increase by about 30°C subjected to aircraft Mach number compared to ambient temperature, is corrected. Similar to Bärfuss et al. (2023), an additional recovery factor dependent on ICH sensor specifics and aircraft Mach number is also applied to account for an incomplete adiabatic process. And the ICH sensors are regularly calibrated under cruise temperature conditions in the atmospheric simulation chamber against a dew/frost point mirror. The overall error introduced by the temperature corrections is about 0.1-0.15 °C.

- 16: include vertical lines for better overview, e.g. for begin of climb?

  Thank you for the suggestion. We included a box indicating the time interval of the climb to the next level.

Minor details:

- 24: according to THE World Meteorological Organization
  Correct. We changed it accordingly

- aircraft is also aircraft in the plural form, please adapt throughout the manuscript, e.g. l.40, 54
  Correct. We changed it accordingly

- put references in chronological order, e.g. l. 51/52, 406
  Thank you for the hint. We changed it accordingly

- 75: deploy IT during...
  Correct. We changed it accordingly

- 125: modificationS
  Correct. We changed it accordingly

- caption of Fig. 3: bracket missing
  Correct. We changed it accordingly

- 185: ThereforeE
  Correct. We changed it accordingly

- 231 thermalLy isolated
  Correct. We changed it accordingly

- 245: in Section 4.4
  Correct. We changed it accordingly

- 246: instrument output frequency of the ozone instrument
  We changed this part of the sentence as follows: "...the output frequency of the ozone instrument..."

- explain all acronyms, e.g. l. 263 NIST
  We added the explanation for NIST (National Institute of Standards and Technology)

- use „laboratory" instead of „lab" throughtout the text, e.g. l. 328
  Correct. We changed it accordingly

- 344 AT a distance
  Correct. We changed it accordingly

- 373: brackets
  Correct. We changed it accordingly

- change order of Fig. 15 and 16, as mentioned in text?

- 482 dot missing
  Correct. We changed it accordingly

- 483 2 dots
  Correct. We changed it accordingly

- 500: AT two altitudes
  Correct. We changed it accordingly

- 504/505: rephrase

  We rephrased this part as follows:

In addition to trace species measurements, two UHSAS instruments deployed on TPC-TOSS and the Learjet provide, for the first time, the opportunity to study the impact of small-scale dynamical features on aerosol concentration and size distribution in the UTLS. A recent study by Joppe et al. (2025) further exploited the potential temperature gradient derived from the dual-platform measurements to analyze the radiative impact of biomass-burning aerosol transported into the UTLS by warm conveyor belt transport.

- 524: all authors
  Correct. We changed it accordingly

References:

Neis, P., Smit, H. G. J., Rohs, S., Bundke, U., Krämer, M., Spelten, N., Ebert, V., Buchholz, B., Thomas, K., and Petzold, A.: Quality assessment of MOZAIC and IAGOS capacitive hygrometers: insights from airborne field studies, Tellus B: Chemical and Physical Meteorology, 67, https://doi.org/10.3402/tellusb.v67.28320, 2015.

Rolf, C., Rohs, S., Smit, H. G. J., Krämer, M., Bozóki, Z., Hofmann, S., Franke, H., Maser, R., Hoor, P., and Petzold, A.: Evaluation of compact hygrometers for continuous airborne measurements, Meteorologische Zeitschrift, 15–34, https://doi.org/10.1127/metz/2023/1187, 2024.

---

## Author Response (AR2)

**EGUSPHERE-2025-3175**

Review of the manuscript "The TropoPause Composition TOwed Sensor Shuttle (TPC-TOSS): A new airborne dual platform approach for atmospheric composition measurements at the tropopause"

by Bozem et al.

Reply to editor

We appreciate further suggestions to improve the manuscript. We will answer to all comments of the editor below point by point. Editor comments are given in standard, answers in red, and changes to the manuscript in blue font.

The authors could resolve all questions by the reviewers and did adjust the manuscript accordingly.

This makes the manuscript almost ready for publication. To finalize, I have a short list of suggestions that came up when reading the revision. These may provide some improvements. The authors hopefully agree.

L 41: Instead of HALO-(AC)3 better use the ACLOUD campaign as a reference. During HALO-AC3 also the HALO aircraft was operated and flew in the tropopause for some time. That's why your statement on the Polar 5/6 operation might be misleading. In the ACLOUD data paper, a discussion on the collocation of P5/P6 is given (https://doi.org/10.5194/essd-11-1853-2019).

Thank you very much for this suggestion. We changed the reference accordingly and rephrased L41 ff.:

One approach in former studies was to perform co-located measurements with two aircraft, for example during CRYSTAL-FACE (Cirrus Regional Study of Tropical Anvils and Cirrus Layers – Florida Area Cirrus Experiment) (Jensen et al., 2004) or with the *Polar 5* and *Polar 6* aircraft during ACLOUD (Ehrlich et al., 2019). The *Polar* aircraft are almost identical making them well suited for co-located measurements as shown by Maherndl et al., 2024. However, flights with *Polar 5* and *Polar 6* aircraft could only be performed below 5 km altitude due to aircraft performance capabilities. In general, coordinated measurements involving two different aircraft often suffer from difficulties of exact horizontal co-location at two different altitudes due to different aircraft speeds, as pointed out by Klingebiel et al. (2017, and references therein). Further, measurements at a vertical distance of just 100-200m are difficult to realize with two aircraft for safety reasons.

L41: The sentence on Polar 5/6 can also be misleading because readers might not know, that the two aircraft are twins with the same performance. The second part of the sentence may imply, that the collocated approach never worked at all. However, with Polar 5/6 being almost identical, the collocated approach worked quite well for some

applications. See Maherndl et al. 2024 (https://acp.copernicus.org/articles/24/13935/2024/)

See rephrased text above.

L56: The introduction ends a little plain. I'm missing a motivation why TPC-TOSS is needed. Any discussion on the requirements to study tropopause gradients?

- What vertical resolution do you need to study tropopause gradients? Or separation between the two payloads?
- What accuracies of tropopause measurements is needed to study the gradients/processes?

In this respect, what is the intended advantage of TPC-TOSS compared to previous approaches. Is the motivation to have the identical instrumentation in two altitudes at the same time?

**We added a paragraph to the introduction addressing the suggestions above:**

The new setup of the TPC-TOSS (TropoPause Composition TOwed Sensor Shuttle) includes measurements of ozone, GPS information, aerosol size distribution from 95-1000 nm as well as sensors for humidity and temperature, which are operated simultaneously on the Learjet.

This approach addresses the challenge of characterizing transient fine-scale structures and composition variability in the tropopause region. Such features are particularly associated with composition gradients and variations in temperature, humidity and ozone. To resolve these structures, simultaneous measurements typically require a vertical resolution of 100-150 m and accuracies better than about 0.2 K for temperature and a few percent for humidity and trace species measurements.

Notably, conventional in situ single-platform approaches cannot provide truly simultaneous observations of transient structures with a lifetime of less than a few minutes. The dual-platform concept combining TPC-TOSS and the Learjet directly addresses this gap by deploying two synchronized payloads at slightly different altitudes. This configuration provides co-located measurements within the tropopause region, minimizing calibration offsets and temporal mismatches. A vertical separation of 100-150 m allows instantaneous determination of gradients and mixing signatures that would otherwise be obscured by sequential profiling. Compared with previous approaches, TPC-TOSS thus offers a unique capability to quantify small-scale transport and mixing processes at the tropopause and to relate observed gradients to underlying dynamical mechanisms.

In the following sections we will present the new setup and will provide uncertainties and individual tests, as well as some examples demonstrating the agreement between

the two platforms. Additionally, we will showcase typical results achieved during the first field setup.

L108: I would further emphasize that you demonstrate here the potential of the identical payloads in TPC-TOSS and the learjet. I would also explicitly state, that for all other instruments references are given (instruments that were operated on one platform only). Readers might have the question: why only describing this selection of instruments and not all. That's why the tandem analysis should be highlighted. Additionally, you may motivate what is needed to prove that the duplicated instruments can be analyzed jointly: comparability, cross calibration, identical performance of the twin instruments must be given and demonstrated as in your study.

**Thank you very much for the suggestion. We modified L108 ff. as follows:**

This synchronized payload with partly identical instrumentation on TPC-TOSS and Learjet in particular allows for the determination of gradients of different quantities which in turn are used to study the effect of small-scale transient features on the UTLS composition. Therefore, the focus of this paper is on the TPC-TOSS and the duplicated instrumentation on the Learjet as part of the dual platform approach. For all other instrumentation on the Learjet and in the underwing pod we provide references for characterization and application of the respective instrument in Table 2 and Table 3. In Sect. 4 we will describe the TPC-TOSS instrumentation in detail also demonstrating the comparability and similar performance of the instruments based on cross calibration in the laboratory, during the intensive operation period on ground and also during research flights. This is in particular essential to ensure that observed differences between the two payloads reflect true atmospheric gradients rather than instrumental offsets.

Table 2-4: All acronyms need to be explained. Maybe in the table caption or a footnote.

We added the explanation of the acronyms to the respective tables.

Fig. 2: I can not identify any instrument on this picture. I only see a metal frame and some cables. This makes the image useless. If there is any reason why you show this images, then make it clearer to the reader, use labels, add a scale, etc.. or simple remove it.

We removed Fig. 2.